# Dictionary Contrastive Learning for Efficient Local Supervision without Auxiliary networks

**Suhwan Choi**[1,2*] **Myeongho Jeon**[1] **Yeonjung Hwang**[1] **Jeonglyul Oh**[1*] **Sungjun Lim**[1*]
**Joonseok Lee**[1,3†] **Myungjoo Kang**[1*†]
[1]Seoul National University, [2]CRABs.ai, [3]Google Research
{schoi828, andyjeon, hyjok, jamesoh0813, lsjung567,
joonseok, mkang}@snu.ac.kr

## Abstract

While backpropagation (BP) has achieved widespread success in deep learning, it faces two prominent challenges: computational inefficiency and biological implausibility. In response to these challenges, local supervision, encompassing Local Learning (LL) and Forward Learning (FL), has emerged as a promising research direction. LL employs module-wise BP to achieve competitive results yet relies on module-wise auxiliary networks, which increase memory and parameter demands. Conversely, FL updates layer weights without BP and auxiliary networks but falls short of BP's performance. This paper proposes a simple yet effective objective within a contrastive learning framework for local supervision without auxiliary networks. Given the insight that the existing contrastive learning framework for local supervision is susceptible to task-irrelevant information without auxiliary networks, we present DICTIONARY CONTRASTIVE LEARNING (DCL) that optimizes the similarity between local features and label embeddings. Our method using static label embeddings yields substantial performance improvements in the FL scenario, outperforming state-of-the-art FL approaches. Moreover, our method using adaptive label embeddings closely approaches the performance achieved by LL while achieving superior memory and parameter efficiency.

## 1 Introduction

Backpropagation (BP) (Rumelhart et al., 1986) has been a fundamental tool in deep learning. However, BP exhibits two inherent limitations. Firstly, the requirement for weight symmetry during forward and backward passes renders BP biologically implausible (Liao et al., 2016). While the causal relationship between biological fidelity and the effectiveness of learning algorithms has yet to be established clearly, numerous deep learning studies have been focused on emulating human biological and cognitive processes (Fei et al., 2022; Taniguchi et al., 2022). Secondly, forward passes can initiate only when backward passes are fully completed (backward locking), and the same applies in reverse (forward locking), which results in computational inefficiencies due to limited parallelization. Furthermore, because weight gradient computation requires storing local activations of each layer, memory usage is also inefficient.

In response, several alternatives to BP have been presented. Feedback alignment (FA) (Lillicrap et al., 2014; 2016) substitutes symmetric feedback weights with fixed random weights. However, it remains constrained by forward/backward locking. Its successor, Direct Feedback Alignment (DFA) (Nøkland, 2016), directly propagates error signals to each layer to alleviate backward locking. Yet, DFA does not resolve the forward locking problem. For this, local supervision leverages local weight updates by minimizing local losses. Specifically, local learning (LL) (Nøkland & Eidnes, 2019; Belilovsky et al., 2020) employs local BP with module-wise auxiliary networks, which process local outputs to align with the targets for local loss computation. Auxiliary networks allow existing LL to achieve performance comparable to BP, but using them at every module substantially increases model parameters.

---

[*]IPAI (Interdisciplinary Program in Artificial Intelligence, Seoul National University)
[†]Corresponding authors

A newer local supervision approach takes a bold leap by entirely eliminating BP and auxiliary networks from LL. In this paper, we refer to this method as forward learning (FL). In FL, weight updates for each layer are guided by layer-specific local losses, avoiding forward/backward locking issues and leading to substantial improvements in computational efficiency. In the absence of BP and auxiliary networks, the essential aspect of implementing FL lies in formulating local targets for loss computation. For instance, the Forward-Forward algorithm (FF) (Hinton, 2022) defines local targets by overlaying one-hot encoded labels onto images, treating them as individual pixels within the image. Since the local outputs include the target information, FF-based methods (Ororbia & Mali, 2023; Lee & Song, 2023) optimize the self-dot product of local outputs for contrastive learning objectives. However, this makes contrastive learning in the FL scenario susceptible to task-irrelevant information in the local outputs, resulting in subpar performance compared to BP and LL.

Our investigation suggests that auxiliary networks play a crucial role in mitigating the impact of task-irrelevant information. In response to the challenges posed by the absence of auxiliary networks, we propose a straightforward yet effective local contrastive learning objective, DICTIONARY CONTRASTIVE LEARNING (DCL), which effectively aligns local outputs with label embedding vectors. We evaluate two versions of DCL: one employing static label embedding vectors tailored for the FL scenario and another featuring adaptive label embedding vectors. Remarkably, our static method significantly outperforms state-of-the-art FL baselines in the FL scenario by discarding task-irrelevant information more effectively. Moreover, our adaptive method showcases performance on par with BP and LL while markedly surpassing LL in terms of parameter and memory efficiency. Further extensive analyses not only support our intuitions and the effectiveness of our method but also unveil intriguing properties of label embedding vectors.

## 2 RELATED WORK

### 2.1 LOCAL LEARNING

The current research trend encompasses two crucial directions in the field of LL. Nøkland & Eidnes (2019) stands as one of the pioneering studies to illustrate that a non-BP learning algorithm can surpass the performance of BP. They utilize two distinct local losses, each originating from a separate auxiliary network pathway. The reconstruction loss measures the L2 distance between the self-similarity matrix of one-hot encoded labels and the local features, while the label prediction loss employs cross-entropy, commonly used in both BP and LL's local layers (Belilovsky et al., 2019; 2020; Pathak et al., 2022). Nøkland & Eidnes (2019) also introduce biologically plausible, BP-free versions of these losses, *i.e.*, LL-bpf. The reconstruction loss uses a standard deviation operation instead of a convolutional auxiliary network, and the label prediction loss employs feedback alignment for weight updates.

In a different vein, Wang et al. (2020) conduct an information theory-based analysis of LL. The authors investigate the influence of the local loss objective on mutual information metrics: $I(\boldsymbol{h}, \boldsymbol{x})$ (between local features and inputs) and $I(\boldsymbol{h}, y)$ (between local features and labels). Compared to BP, $I(\boldsymbol{h}, \boldsymbol{x})$ and $I(\boldsymbol{h}, y)$ exhibit more pronounced decreases as the LL's layers deepen. To mitigate the loss of $I(\boldsymbol{h}, \boldsymbol{x})$ during the forward passes, the authors introduce a module-wise reconstruction loss, which encourages local features to preserve input information. Furthermore, they prove that minimizing the supervised contrastive loss in Eq. (1) (Khosla et al., 2020) maximizes the lower bound of $I(\boldsymbol{h}, y)$. Utilizing both contrastive and reconstruction losses leads to significant performance improvements, although it comes at the cost of increased computational demands stemming from the module-wise reconstruction loss.

### 2.2 FORWARD LEARNING

As FL aims to avoid reliance on BP, it limits the use of auxiliary networks, which guide features toward their targets with local updates by BP. In the absence of these auxiliary networks, the formulation of local targets becomes crucial. To address this challenge, the Forward Forward Algorithm (FF) (Hinton, 2022) defines local targets by overlaying one-hot encoded labels onto images, treating them as individual pixels within the image. Employing Noise Contrastive Estimation (NCE) (Gutmann & Hyvärinen, 2010), FF optimizes the self-dot product of local features to be above a certain threshold if the overlaid labels match the images (positive pair). Otherwise (negative pair), local features are

optimized to be under a certain threshold. Other FF-based approaches follow the same NCE objective and label overlay, with specialized architecture (PFF) (Ororbia & Mali, 2023) or refined label overlay technique (SymBa) (Lee & Song, 2023).

Taking a distinct path apart from FF, the cascaded forward algorithm (CaFo) (Zhao et al., 2023) linearly projects flattened local features to local label prediction. To achieve this without BP, CaFo freezes the weights of feedforward layers, reserving weight updates solely for the local linear projection layers. On the other hand, the direct random target projection (DRPT) (Frenkel et al., 2021) treats one-hot encoded labels as error signals themselves and uses fixed random weights to propagate the error signals to each layer. As one-hot encoded labels are locally accessible, parameter updates can occur during each forward pass.

## 3 BACKGROUND

Contrastive learning stands as a powerful tool for representation learning, and its efficacy has also been demonstrated in the context of LL and FL. InfoPro (Wang et al., 2020), an LL method, compares local features derived from module-wise auxiliary networks. In contrast, FF-based approaches leverage self-dot products of local features, as the features contain the label information. In Wang et al. (2020), for a batch of local outputs $\boldsymbol{h} \in \mathbb{R}^{C \times H \times W}$ from a forward pass layer, the local contrastive loss is defined as follows:

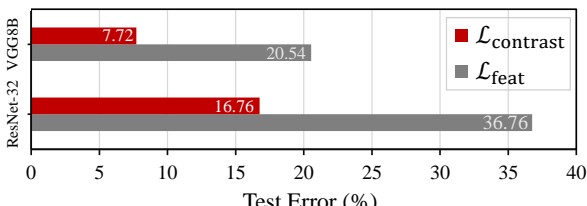

Figure 1: Comparison of $\mathcal{L}_{\text{feat}}$ with $\mathcal{L}_{\text{contrast}}$ on CIFAR-10. Both losses employ Eq. (1), but $\mathcal{L}_{\text{feat}}$ utilizes no auxiliary network, such that $f_\phi(\boldsymbol{h}) = \boldsymbol{h}$. Training details are available in Appendix J.2.4.

$$\mathcal{L}_{\text{contrast}} = -\frac{1}{\sum_{i \neq j} \mathbf{1}_{y_i = y_j}} \sum_{i \neq j} \left[ \mathbf{1}_{y_i = y_j} \log \frac{\exp\left(\boldsymbol{a}_i^\top \boldsymbol{a}_j / \tau\right)}{\sum_{k=1}^N \mathbf{1}_{i \neq k} \exp\left(\boldsymbol{a}_i^\top \boldsymbol{a}_k / \tau\right)} \right], \quad \boldsymbol{a}_i = f_\phi\left(\boldsymbol{h}_i\right), \quad (1)$$

where $\tau$ is a temperature hyperparameter, $y \in \{1, ..., Z\}$ is the ground truth label, and $f_\phi$ is an auxiliary network. In Eq. (1), $\boldsymbol{a}_i$ and $\boldsymbol{a}_j$ are positive features, such that $y_i = y_j$. This function aims to maximize the similarity between positive features while minimizing that between negative features. When the auxiliary network $f_\phi$ is an identity function, this objective represents the FL scenario. For convenience, we use $\mathcal{L}_{\text{feat}}$ to denote $\mathcal{L}_{\text{contrast}}$ with $f_\phi(\boldsymbol{h}) = \boldsymbol{h}$. Please note that the primary objective of this article is to enhance the performance with contrastive learning in the absence of auxiliary networks. To that end, $\mathcal{L}_{\text{feat}}$ can be regarded as the foundational framework that will be further elaborated upon in the following sections.

Although both FF-based approaches and InfoPro exploit concepts of contrastive learning to formulate local objectives, the performance of FF-based approaches falls short compared to InfoPro (LL). Additionally, with precisely the same setup, we compare the performance of $\mathcal{L}_{\text{contrast}}$ and $\mathcal{L}_{\text{feat}}$ and report the significant performance gap in Figure 1. These findings underscore the significance of auxiliary networks in local contrastive learning, setting the stage for our goal to develop a local contrastive learning framework that excels without auxiliary networks.

## 4 METHODOLOGY

### 4.1 MOTIVATION

To enhance the model's performance using local contrastive learning without auxiliary networks, we commence our method design by examining the role of auxiliary networks. We suggest that the notable disparity in performance between $\mathcal{L}_{\text{contrast}}$ and $\mathcal{L}_{\text{feat}}$ can be attributed to the presence of the mutual information $I(\boldsymbol{h}, r)$, where $r$, referred to as a nuisance, denotes a task-irrelevant variable in $\boldsymbol{x}$. Then, given a task-relevant variable $y$, it follows that $I(r, y) = 0$ because mutual information $I$ signifies the amount of information obtained about one random variable by observing another (Achille & Soatto, 2018). $\mathcal{L}_{\text{feat}}$ maximizes similarity between local features ($\boldsymbol{h}^{+\top} \boldsymbol{h}^p$), rather than similarity between $\boldsymbol{h}$ and labels $y$. Accordingly, maximizing the similarity between local features could also

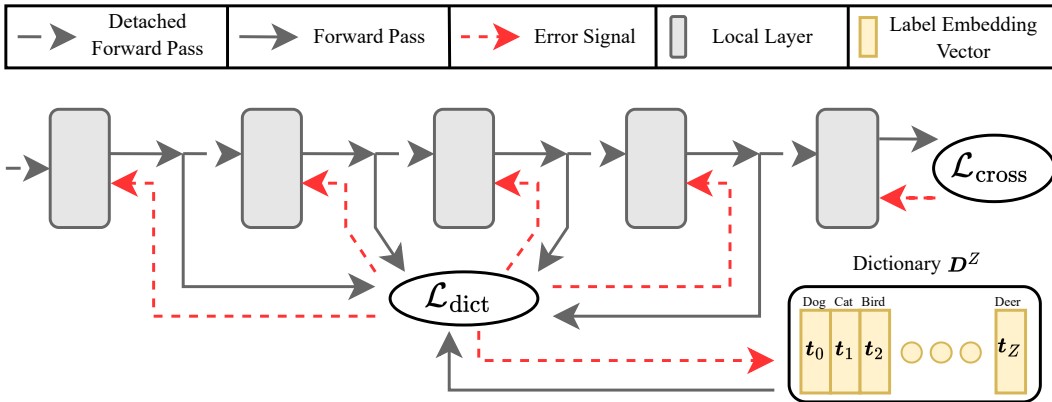

Figure 2: Overview of DCL. Every input is detached before a forward pass, ensuring that no layer propagates error signals backward. The final layer $f_L$ receives its error signal from the cross-entropy loss $\mathcal{L}_{\text{cross}}$. Every other layer receives its error signal from $\mathcal{L}_{\text{dict}}$, which optimizes the similarity between layer-wise local features and label embedding vectors.

lead to an increase in $I(r^+, r^p)$, misleading the model to consider task-irrelevant information as meaningful features.

In this respect, auxiliary networks have the capacity to filter out $r$, reducing the impact of $r$ in LL (see Appendix A for more details). However, in FL where auxiliary networks are unavailable, the influence of $r$ becomes more detrimental and noticeable. This likely explains the subpar performance of existing contrastive learning in the FL scenario.

### 4.2 DICTIONARY CONTRASTIVE LOSS

To address the problem with $r$ in FL, we propose a novel objective that directly maximizes the similarity between $h$ and embedding vectors corresponding to target labels.

**Mapping labels to embedding vectors.**    To obtain label embedding $\mathbf{t}_z$ from each target label $y_z$, we define an embedding mapping function $f_m$. The embedding mapping function $f_m : \mathbb{N} \to \mathbb{R}^{C_D}$ is a one-to-one mapping from a label to a $C_D$-dimensional label embedding vector, which can be directly compared with dense local features. Every label embedding vector $\mathbf{t}$ is initialized as the standard normal random vector, each element of which is i.i.d. random variable sampled from the standard normal distribution. For $Z$ label classes, we have a label embedding dictionary $\boldsymbol{D}^Z = \{f_m(y_z) \mid y_z \in \{1, ..., Z\}\}$, where $f_m(y_z) = \boldsymbol{t}_z$.

**Local features.**    We aim to optimize the similarity between label embedding vectors $\boldsymbol{t}$ and local features $\boldsymbol{h}$. First of all, because the shapes of local features may vary across different architectures, we standardize the representation of $\boldsymbol{h}$. We represent local features at the l-th layer as $\boldsymbol{h}_l \in \mathbb{R}^{C_h^l \times K_l}$, where $K_l$ is the number of $C_h^l$ dimensional feature vectors. Because $C_h^l$ can differ for each layer $l$, we define the label embedding vector dimension $C_D$ as $\max C_h^l$. For fully connected layers (FC), we reshape a flat output vector $\boldsymbol{h}_{\text{flat}} \in \mathbb{R}^{C_h^l K_l}$ into $\boldsymbol{h_l} \in \mathbb{R}^{C_h^l \times K_l}$ (See Appendix E for experiments concerning the optimal selection of $K$). For convolutional layers, local outputs are feature maps $\boldsymbol{h}_l \in \mathbb{R}^{C_h^l \times H_l \times W_l}$, where $C_h^l$ signifies the channel dimension, whereas $H_l$ and $W_l$ denote the feature maps' height and width, respectively. We maintain the integrity of the $C_h^l$-dimensional vectors within the feature maps by setting $K_l = H_l W_l$. To prevent BP across layers, we employ the stop gradient operator $sg[\cdot]$, such that $\boldsymbol{h}_l = f_l(sg[\boldsymbol{h}_{l-1}])$.

**Training objective.**    The weights of the final prediction layer $f_L$ are updated through the standard cross-entropy loss. All other layers $f_l$ for $l = 1, ..., L-1$ leverage the dictionary contrastive loss

$\mathcal{L}_{\text{dict}}$ to update their weights. For a batch of local features $\{\boldsymbol{h}_n\}_{n=1}^N$, we minimize the loss:

$$\mathcal{L}_{\text{dict}} = -\frac{1}{N} \sum_{n=1}^N \left[ \log \frac{\exp(\langle \bar{\boldsymbol{h}}_n, \boldsymbol{t}'_+ \rangle)}{\sum_{z=1}^Z \exp(\langle \bar{\boldsymbol{h}}_n, \boldsymbol{t}'_z \rangle)} \right], \ \boldsymbol{t}' \in \{pool_l(\boldsymbol{t}_z) | \ \boldsymbol{t}_z \in \boldsymbol{D}^Z \}, \qquad (2)$$

where we define $\bar{\boldsymbol{h}}_n := \frac{1}{K} \sum_{k=1}^K \boldsymbol{h}_n^k$ [1], $\langle \cdot, \cdot \rangle$ denotes the dot product, and the label embedding vector $\boldsymbol{t}_+$ corresponds to the label of $\boldsymbol{h}_n$. The dimension of local feature vectors may vary across different layers $l$. To align the vector dimension of $\boldsymbol{t}_z \in \mathbb{R}^{C_D}$ to that of $\bar{\boldsymbol{h}} \in \mathbb{R}^{C_h^l}$, we employ the one-dimensional average pooling $pool_l : \mathbb{R}^{C_D} \to \mathbb{R}^{C_h^l}$ differently for each layer $l$.

In this paper, we apply static label embedding vectors solely in the FL scenario to address the constraint that layer weights and label embedding weights cannot be updated simultaneously. Technically, this limitation makes our adaptive method partially forward/backward locked, although this locking is limited in scope and negligible in practice. Nonetheless, for a fair evaluation against other FL approaches, which are fully forward/backward unlocked, our static method maintains initial label embedding weights constant throughout the training process, such that $\boldsymbol{t}_z^{static} = sg[\boldsymbol{t}_z]$. In contrast, in scenarios not bound by the FL-specific constraint, label embedding vectors are adaptive, updating their weights at every layer via error signals from $\mathcal{L}_{\text{dict}}$. Figure 2 illustrates the training workflow using $\mathcal{L}_{\text{dict}}$. We establish that the minimization of $\mathcal{L}_{\text{dict}}$ maximizes a lower bound of $I(\boldsymbol{h}, y)$ in Appendix B.

**Comparison with other contrastive objectives.** Contrastive objectives based on InfoNCE (Oord et al., 2018) are known for their sensitivity to the size of negative samples (Khosla et al., 2020; Radford et al., 2021; Chen et al., 2020). These contrastive objectives, including $\mathcal{L}_{\text{feat}}$, often utilize in-batches negative samples and tend to show improved performance with larger batch sizes $N$ (Wang et al., 2020; Lee et al., 2018). In contrast, the number of negative samples in $\mathcal{L}_{\text{dict}}$ corresponds to $Z - 1$. Hence, the efficacy of $\mathcal{L}_{\text{dict}}$ depends on the number of classes. Empirical results confirm that a higher number of label classes $Z$ tends to yield more pronounced performance when compared to using static label embedding vectors. Nevertheless, competitive performance is still achieved even with fewer classes.

**Layer-wise prediction.** Minimizing $\mathcal{L}_{\text{dict}}$ maximizes the similarity between local features $\boldsymbol{h}$ and their corresponding label embedding vectors $\boldsymbol{t}_+$, while concurrently minimizing the similarity to non-corresponding label embedding vectors. Leveraging this property of $\mathcal{L}_{\text{dict}}$, $\boldsymbol{D}^Z$ can be employed for inference without the final linear classifier $f_L$. Predictions can be generated by selecting the target label with the highest similarity to the feature vectors:

$$\hat{y} = \arg\max_z \langle \bar{\boldsymbol{h}}, \boldsymbol{t}'_z \rangle, \ \boldsymbol{t}' \in \{pool_l(\boldsymbol{t}_z) | \boldsymbol{t}_z \in \boldsymbol{D}^Z \}. \qquad (3)$$

Accordingly, prediction is possible at every layer. Furthermore, this allows for a weighted sum of layer-wise predictions to serve as the global prediction as in Belilovsky et al. (2019); Zhao et al. (2023). This approach surpasses predictions made solely by $f_L$. Experiments on layer-wise prediction are available in Appendix C.

## 5 EXPERIMENTS

### 5.1 EXPERIMENTAL SETUPS

Due to the substantial performance disparities between FL and LL, we conduct separate comparisons. In the FL scenario, our static method `DCL-S`, featuring static label embeddings, is evaluated using simple fully connected (FC) and convolutional (Conv) architectures in line with the FL baselines. For our adaptive method `DCL`, we employ the VGG8B (Simonyan & Zisserman, 2015) architecture as utilized by Nøkland & Eidnes (2019).

---

[1] In practice, we opt for $\frac{1}{K} \sum_{k=1}^K \langle \boldsymbol{h}_n^k, \boldsymbol{t} \rangle$ over $\langle \bar{\boldsymbol{h}}_n, \boldsymbol{t} \rangle$. While both are mathematically equivalent given the linearity of the dot product, averaging feature vectors first leads to performance degradation due to a greater loss in floating-point precision.

Table 1: Test errors and the number of parameters with convolutional networks. We highlight the top-performing results in bold and the second-best results by underlining them. Results marked with an asterisk (∗) indicate replicated results.

| | Approach | MNIST | | CIFAR-10 | | CIFAR-100 | |
|---|---|---|---|---|---|---|---|
| | | *Params.* | *Err.* | *Params.* | *Err.* | *Params.* | *Err.* |
| | **BP** | 152K | 2.63 | 153K | **22.84** | 1.43M | 46.41 |
| FL | CaFo | 243K | **1.20** | 243K | 32.57 | 2.4M | 59.24 |
| | DRTP | 1.8M | 1.48 | 4.1M | 31.04 | 19.2M | *65.02 |
| | $\mathcal{L}_{\text{feat}}$ | 152K | 11.55 | 153K | 43.67 | 1.43M | 67.69 |
| | DCL-S | 152K | 3.21 | 153K | 25.86 | 1.43M | 54.21 |

Table 2: Test errors and the number of parameters with fully connected networks. The same architecture is used except for PFF and DRTP.

| | Approach | MNIST | | CIFAR-10 | | CIFAR-100 | |
|---|---|---|---|---|---|---|---|
| | | *Params.* | *Err.* | *Params.* | *Err.* | *Params.* | *Err.* |
| | **BP** | 1.87M | **1.29** | 18.9M | 34.73 | 19.2M | 65.94 |
| FL | FF | 1.87M | 1.36 | 18.9M | 41.00 | 19.2M | *95.20 |
| | PFF | 23.0M | 1.34 | 32.4M | *50.14 | 32.7M | *81.31 |
| | DRTP | 6.3M | 4.00 | 16.5M | 51.27 | 46.4M | *88.32 |
| | SymBa | 5.6M | 1.42 | 18.9M | 40.91 | 19.2M | 70.72 |
| | $\mathcal{L}_{\text{feat}}$ | 1.87M | 3.13 | 18.9M | 43.29 | 19.2M | 70.67 |
| | DCL | 1.87M | 1.46 | 18.9M | 35.12 | 19.2M | 66.48 |

Table 3: Test errors across different datasets using the VGG8B architecture employed by Nøkland & Eidnes (2019). LL-predsim denotes an LL model trained with both prediction (LL-pred) and similarity-matching loss (LL-sim). LL-bpf, the BP-free version of LL-predsim, also utilizes two local losses explained in Section 2.1. LL-contrec signifies an LL model trained with both $\mathcal{L}_{\text{contrast}}$ in Eq. (1) (LL-cont) and image reconstruction loss used by Wang et al. (2020), as explained in Section 2.1. The top-performing results are highlighted in bold. The second-best results are underlined.

| Loss Type | Method | MNIST | F-MNIST | CIFAR-10 | CIFAR-100 | SVHN | STL-10 |
|---|---|---|---|---|---|---|---|
| Single Global Loss | BP | **0.26** | **4.53** | 5.99 | 26.20 | 2.29 | 33.08 |
| Two Local Losses | LL-contrec | *0.65 | *5.71 | *9.02 | *31.35 | *2.34 | *29.74 |
| | LL-predsim | 0.31 | 4.65 | **5.58** | **24.10** | **1.74** | **20.51** |
| | LL-bpf | *0.35 | *5.68 | 9.02 | *37.60 | *2.31 | *26.12 |
| Single Local Loss | LL-cont | *0.37 | *5.92 | *7.72 | *31.19 | *2.29 | *26.83 |
| | LL-pred | 0.40 | 5.66 | 8.40 | 29.30 | 2.12 | 26.83 |
| | LL-sim | 0.65 | 5.12 | 7.16 | 32.60 | 1.89 | 23.15 |
| Single Local Loss | DCL | 0.33 | 5.52 | 8.64 | 31.75 | 2.19 | 22.87 |

**Experiments with FC and Conv architectures.** We compare our method against FL approaches, including $\mathcal{L}_{\text{feat}}$, as well as the standard BP. These comparisons are carried out on MNIST (LeCun, 1998), CIFAR-10, and CIFAR-100 (Krizhevsky et al., 2009). We benchmark our method against FL models: PFF (Ororbia & Mali, 2023), DRTP (Frenkel et al., 2021) FF (Hinton, 2022), and SymBa (Lee & Song, 2023). Our FC models share the same FC architecture as FF and SymBa for a fair comparison. We also evaluate against Conv models: DRTP and CaFo (Zhao et al., 2023). Since DRTP and CaFo use different architectures, we employ a Conv architecture with fewer parameters. The BP and $\mathcal{L}_{\text{feat}}$ baseline models are trained using the same architectures and hyperparameters as our models. Details on architectures, datasets, and training setups are provided in Appendix J.

**Experiments with the VGG8B architecture.** We employ the VGG8B architecture to compare against LL, BP-free LL (LL-bpf), and BP. Our evaluation encompasses the same datasets employed by Nøkland & Eidnes (2019): MNIST, CIFAR-10, CIFAR-100, Fashion-MNIST (Xiao et al., 2017), SVHN (Netzer et al., 2011), and STL-10 (Coates et al., 2011). We use the same batch size, learning rate, learning rate schedule, and training epochs used by Nøkland & Eidnes (2019). Please refer to Appendix J for more details.

## 5.2 MAIN RESULTS

**Comparison with FL approaches.** We first compare our static method against BP and other FL approaches. Table 1 and Table 2 report the test error and number of parameters on MNIST, CIFAR-10, and CIFAR-100. On MNIST, CaFo continues to exhibit the best performance among FL models. However, our objective outshines other FL approaches for more realistic datasets (CIFAR-10 and CIFAR-100). Table 1 shows that our Conv model outperforms other FL models significantly despite having fewer parameters. Our FC models also surpass other FC models on CIFAR-10 and CIFAR-100, as depicted in Table 2. Furthermore, our method consistently outperforms $\mathcal{L}_{\text{feat}}$ across all datasets.

Table 4: Comparison of GPU memory usage and model parameters for VGG8B models. *Memory* denotes the peak GPU memory consumption measured during single GPU training with a batch size of 128. $\Delta\theta$ represents the increase in the number of parameters compared to the BP baseline.

| Method | **MNIST**, **F-MNIST** | | **CIAFR-10**, **SVHN** | | **CIAFR-100** | | **STL-10** | |
| | $\Delta\theta$ | *Memory* | $\Delta\theta$ | *Memory* | $\Delta\theta$ | *Memory* | $\Delta\theta$ | *Memory* |
|---|---|---|---|---|---|---|---|---|
| BP | **0** | 847 MiB | **0** | 1086 MiB | **0** | 1088 MiB | **0** | 2315 MiB |
| LL-contrec | 1.15M | 811 MiB | 2.07M | 1049 MiB | 2.07M | 1050 MiB | 2.07M | 5954 MiB |
| LL-predsim | 9.53M | 1038 MiB | 9.60M | 1291MiB | 10.9M | 1310 MiB | 9.60M | 2594 MiB |
| LL-bpf | 3.28M | 708 MiB | 3.28M | 895 MiB | 3.28M | 897 MiB | 3.28M | 1851 MiB |
| LL-cont | 918K | 695 MiB | 1.84M | 894 MiB | 1.84M | 895 MiB | 1.84M | 1846 MiB |
| LL-pred | 71.8K | 682 MiB | 143K | 870 MiB | 1.43M | 890 MiB | 143K | 1826 MiB |
| LL-sim | 9.46M | 933 MiB | 9.46M | 1154 MiB | 9.46M | 1156 MiB | 9.46M | 2290 MiB |
| DCL | 5.12K | **580 MiB** | 5.12K | **747 MiB** | 51.2K | **751 MiB** | 5.12K | **1589 MiB** |

Overall, for each architecture type, our method demonstrates superior performance and scalability compared to other FL methods.

**Comparison with LL approaches.** We then compare our adaptive method against LL and LL-bpf. Table 3 exhibits the test error across various datasets. Our methods outperform the BP baselines on the SVHN and STL-10 datasets. For each dataset, our models exhibit competitive performance against LL models trained with a single local loss, as indicated in Table 3. While our models generally perform well, LL-predsim, trained with two local loss functions, still outperforms our method. However, across all datasets, our models consistently achieve better results than LL-bpf, the BP-free version of LL-predsim. It is also worth highlighting that auxiliary networks in LL entail a significant increase in the number of parameters. In contrast, our method introduces much fewer additional parameters: $Z \times C_D$. Table 4 highlights that our methods achieve better memory efficiency over both LL and BP and need fewer parameters compared to LL.

## 5.3 FURTHER ANALYSIS AND DISCUSSION

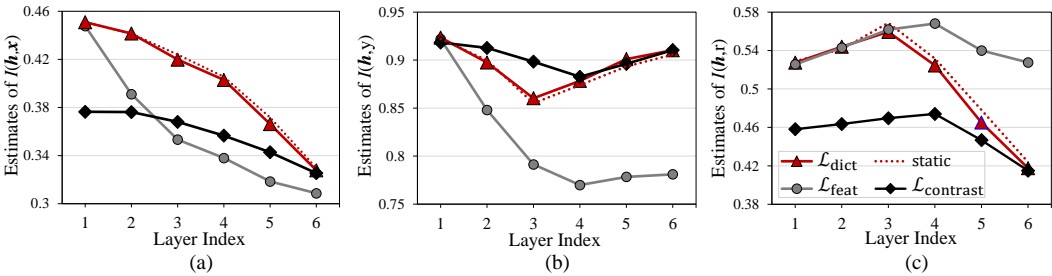

Figure 3: Task-irrelevant information captured by intermediate layers of VGG8B. (a) Estimates of $I(\boldsymbol{h}, \boldsymbol{x})$, (b) Estimates of $I(\boldsymbol{h}, y)$, and (c) Estimates of $I(\boldsymbol{h}, r)$. We train networks to reconstruct input images $\boldsymbol{x}$ from local features $\boldsymbol{h}$, employing the train loss as the estimate of $I(\boldsymbol{h}, \boldsymbol{x})$. Likewise, we train networks to classify labels $y$ from local features $\boldsymbol{h}$ and utilize the accuracy as the estimate of $I(\boldsymbol{h}, y)$. $I(\boldsymbol{h}, \boldsymbol{x}) - I(\boldsymbol{h}, y)$ equals to the upper bound of $I(\boldsymbol{h}, r)$ (Wang et al., 2020), but the substraction yields a negative value because $I_{est}(\boldsymbol{h}, \boldsymbol{x}) < I_{est}(\boldsymbol{h}, y)$. To rectify this, we adjust the scale of the estimated upper bound by adding 1. More details on the mutual information experiments are available in Appendix J.2.7.

**Robustness on task-irrelevant information.** To recast our motivation discussed in Section 4.1, we analyze our objective in comparison to $\mathcal{L}_{\text{feat}}$ and $\mathcal{L}_{\text{contrast}}$ regarding the presence of task-irrelevant information, as depicted in Figure 3. Table 1 and Table 2 highlight the performance gaps between

$\mathcal{L}_{\text{dict}}$ and $\mathcal{L}_{\text{feat}}$, especially with convolutional networks. We interpret these differences through the lens of task-irrelevant information. As our assumption, Figure 3 exhibits that $\mathcal{L}_{\text{feat}}$ encounters the nuisance problem; that is, there is no reduction of $I(\boldsymbol{h}, r)$ from the baseline. In contrast, $\mathcal{L}_{\text{dict}}$ effectively reduces $I(\boldsymbol{h}, r)$ starting from the fourth layer, ultimately matching the $I(\boldsymbol{h}, r)$ level achieved by auxiliary networks ($\mathcal{L}_{\text{contrast}}$), even when label embedding vectors are static. Appendix J.2.7 presents the methods used for estimating mutual information in detail.

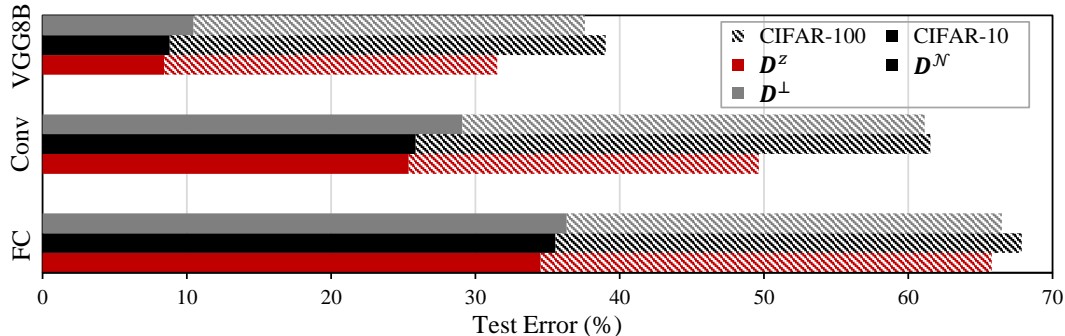

Figure 4: Performance with different types of dictionaries. $\boldsymbol{D}^Z$ is a dictionary of adaptive label embedding vectors. $\boldsymbol{D}^N$ contains static, standard normal random vectors. $\boldsymbol{D}^\perp$ consists of static, orthogonal vectors. Vectors in both $\boldsymbol{D}^N$ and $\boldsymbol{D}^\perp$ have been scaled to match the norm of $\boldsymbol{D}^Z$.

**The effectiveness of adaptive embeddings.** Figure 4 depicts the effectiveness of adaptive embeddings in contrast to static embeddings across CIFAR-10 and CIFAR-100. Models trained with a dictionary of adaptive label embeddings $\boldsymbol{D}^Z$ consistently outperform models trained with a dictionary of static label embedding vectors, regardless of whether the compared static embeddings are random ($\boldsymbol{D}^N$) or orthogonal ($\boldsymbol{D}^\perp$). In particular, the performance gaps for convolutional architectures are more pronounced on CIFAR-100 than on CIFAR-10.

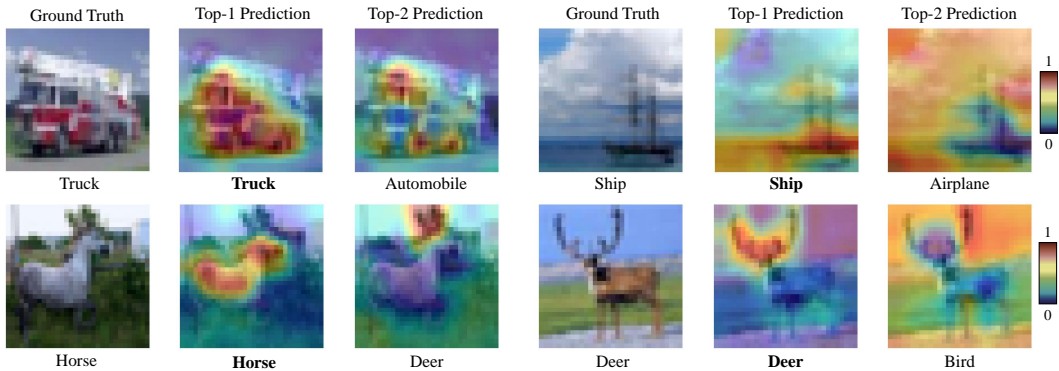

Figure 5: Saliency maps. For a label $y_i$, saliency corresponds to the dot product between the $k$-th feature vector $\boldsymbol{h}^k$ and the embedding vector of $y_i$. The color on the heatmaps represents the level of saliency. For visualization on the heatmaps, saliency is normalized between 0 and 1.

**Explainability with label embeddings.** Figure 5 showcases visualizations of the saliency maps for the two labels with the highest confidence as predicted by the final layer of VGG8B trained on CIFAR-10. These saliency maps are generated through the dot product between the label embedding vector and individual local feature vectors, with each local feature vector representing a distinct region within an image. Regarding the top-1 labels, there is a clear alignment between the saliency maps and the regions within the input images that are relevant to those particular labels. For example, the saliency linked to the top-1 label "Horse" precisely matches the horse's body.

The saliency maps can also provide a convincing explanation for the high confidence of incorrect labels. In Figure 5, the saliency of the top-2 label "Deer" in the "Horse" image suggests that the model

is hallucinating an antler, confused by the two trees near the horse's head. This misinterpretation occurs because antlers often generate elevated saliency for the "Deer" label, as demonstrated by the saliency associated with the top-1 label "Deer" in the corresponding "Deer" image. More examples are available in Appendix K.3.

**Semantic properties of adaptive embeddings.**
Figure 6 illustrates the semantic relationship between adaptive label embedding vectors on CIFAR-100, which comprises 20 super-labels, each encompassing 5 sub-labels. The clustering of these vectors highlights their capacity for semantic learning, akin to their alignment with label-specific salient features in Figure 5. Embeddings from the same super-labels tend to cluster together, while those from semantically similar but different super-labels also show proximity. For example, "forest" is closer to "trees," and "chimpanzee" is closer to "people" than to other embeddings within their super-label groups. Please refer to Appendix D for additional experiments on super-labels.

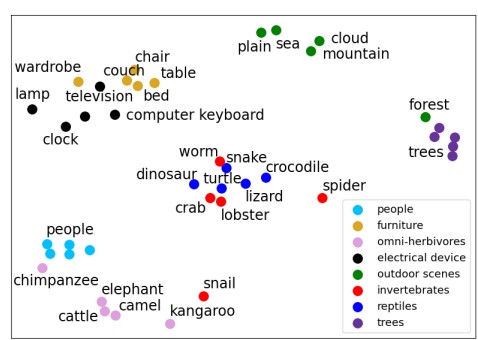

Figure 6: t-SNE of label embeddings. See Appendix K.2 for the complete plot.

**Adaptive label embeddings and confusion rates.**
Figure 7 captures the correlation between the confusion rates and adaptive label embedding similarity on CIFAR-10. With the labels $y_i$ and $y_j$, the confusion rate for $y_i$ and $y_j$ is defined by the average of $\text{conf}_{ij}$ and $\text{conf}_{ji}$. Here, confusion $\text{conf}_{ij}$ is the test error incurred when incorrectly predicting $y_i$ as $y_j$.

Figure 7 illustrates that as the confusion rate between labels increases, the embedding representations of these labels become more distinct. This suggests that when the model struggles to discriminate between two labels, it compensates by dynamically adjusting the label embedding space to facilitate better label separation.

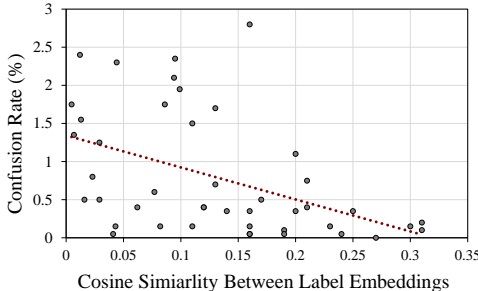

Figure 7: Relationship between confusion rates and label embedding similarity. Each point represents a pair of labels $y_i$ and $y_j$. Please refer to Appendix K.1 for exact values.

**Comparision between average pooling and projection.**
We consider a fully connected layer $f_P^l : \mathbb{R}^{C_D} \to \mathbb{R}^{C_h^l}$ as a substitute for the one-dimensional average pooling $pool_l$ employed in Eq. (2). The linear projection layer $f_P^l$ maps $C_D$-dimensional label embedding vectors to $C_h^l$-dimensional label embedding vectors, such that $f_P^l(\boldsymbol{t}_z) = \boldsymbol{t}_z^l$. Table 5 illustrates that $pool^l$ outperforms $f_P^l$ in performance and memory/parameter efficiency.

Table 5: Projection vs. average pooling.

|  | **CIAFR-10** | | **CIAFR-100** | |
|---|---|---|---|---|
|  | *Err.* | *Memory* | *Err.* | *Memory* |
| $f_P^l$ | 12.47 | 753 MiB | 36.08 | 770 MiB |
| $pool_l$ | **8.64** | **747 MiB** | **31.75** | **751 MiB** |

## 6 CONCLUSION

In this paper, we find that the limited efficacy of conventional contrastive learning objectives without auxiliary networks primarily arises from the inclusion of task-irrelevant information. To address this challenge, we present a novel objective, DCL, which directly aligns local features with label-specific embedding vectors. Even without auxiliary networks, our approach effectively discards task-irrelevant information, outperforming other FL approaches by a large margin. In addition, our method using adaptive label embedding vectors achieves performance levels comparable to those of BP and LL while maintaining superior parameter/memory efficiency compared to LL. Further extensive analyses support our model design and its effectiveness. We aspire for this work to pave a valuable path forward, positioning DCL as a formidable alternative to BP.

ACKNOWLEDGMENT

This work was supported by 1) the NRF grants [2021R1A2C3010887, 2021H1D3A2A03038607, 2022R1C1C1010627, RS-2023-0022663], 2) Artificial intelligence industrial convergence cluster development project funded by the Ministry of Science and ICT (MSIT, Korea) & Gwangju Metropolitan City, 3) Institute of Information & communications Technology Planning & Evaluation (IITP) grant funded by the Korea government (MSIT) [2022-0-00264, 2021-0-01343-004, Artificial Intelligence Graduate School Program (Seoul National University)], and 4) Youlchon Foundation (Nongshim Corporation and affiliated companies). We also extend our sincere thanks to CRABs.ai for their generous financial support and the provision of GPU resources.

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

APPENDIX

## A  THE EFFECTIVENESS OF AUXILIARY NETWORKS IN DISCARDING TASK-IRRELEVANT INFORMATION

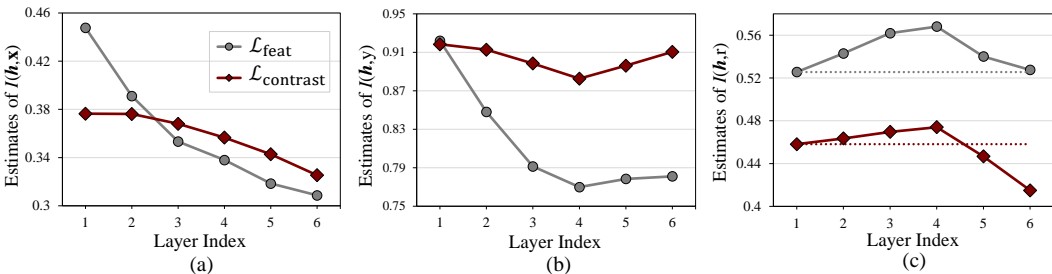

Figure I: Information contents of local features from VGG8B models trained with $\mathcal{L}_{\text{contrast}}$ and $\mathcal{L}_{\text{feat}}$. (a) Estimates of mutual information between local features and inputs $I(\boldsymbol{h}, \boldsymbol{x})$. (b) Estimates of mutual information between local features and labels $I(\boldsymbol{h}, y)$. (c) Estimates of task-irrelevant information in local features $I(\boldsymbol{h}, \boldsymbol{r})$. $I(\boldsymbol{h}, \boldsymbol{x})$ can be estimated by reconstructing $\boldsymbol{x}$ from $\boldsymbol{h}$. $I(\boldsymbol{h}, y)$ can be estimated by classifying $y$ from $\boldsymbol{h}$. We obtain the estimated upper bound of $I(\boldsymbol{h}, \boldsymbol{r})$ by subtracting $I_{est}(\boldsymbol{h}, y)$ from $I_{est}(\boldsymbol{h}, \boldsymbol{x})$. The subtraction results in negative values because the values of $I_{est}(\boldsymbol{h}, \boldsymbol{x})$ are smaller. Thus, we adjust the scale by adding 1. Implementation details are available in Appendix J.2.7.

To demonstrate the effectiveness of auxiliary networks in reducing $I(\boldsymbol{h}, \boldsymbol{r})$, we compare local features of VGG8B models trained with $\mathcal{L}_{\text{contrast}}$ and its auxiliary network-free counterpart, $\mathcal{L}_{\text{feat}}$. Figure I underlines the pronounced decrease of $r$ in local features when auxiliary networks are employed, even from early layers.

## B  DEMONSTRATING THAT MINIMIZATION OF $\mathcal{L}_{\text{dict}}$ MAXIMIZES THE LOWER BOUND OF TASK-RELEVANT INFORMATION

This section demonstrates that minimizing the dictionary contrastive loss,

$$\mathcal{L}_{\text{dict}} = -\frac{1}{N} \sum_{n=1}^{N} \left[ \log \frac{\exp\langle \boldsymbol{h}_n, \boldsymbol{t}_+ \rangle}{\sum_{z=1}^{Z} \exp\langle \boldsymbol{h}_n, \boldsymbol{t}_z \rangle} \right], \ \boldsymbol{h}_n := \frac{1}{K} \sum_{k=1}^{K} \boldsymbol{h}_n^k, \ \boldsymbol{t} \in \boldsymbol{D}^Z, \tag{4}$$

maximizes a lower bound of $I(\boldsymbol{h}, y)$. We follow the progression of the proof by Wang et al. (2020) showing that minimizing $\mathcal{L}_{\text{contrast}}$ in Eq. (1) maximizes a lower bound of mutual information $I(\boldsymbol{h}, y) = \mathbb{E}_{\boldsymbol{h},y} \log \frac{p(\boldsymbol{h},y)}{p(\boldsymbol{h})p(y)}$.

Suppose that we sample a local feature $\boldsymbol{h}_n$ from a set of $N$ local features $\boldsymbol{X} = \{\boldsymbol{h}_1, ..., \boldsymbol{h}_N\}$. Given that $\boldsymbol{h}_n$ corresponds to a label $y_z$, $\boldsymbol{h}_n$ also corresponds to a label embedding vector $\boldsymbol{t}_z$ because we have $f_m(y_z) = \boldsymbol{t}_z$, where the embedding mapping function $f_m$ is the one-to-one mapping. We use $\boldsymbol{t}_+$ to denote such positive label embedding vector. For $Z$ label classes, we have a label embedding dictionary $\boldsymbol{D}^Z = \boldsymbol{t}_+ \cup \boldsymbol{D}_{\text{neg}}$, where $\boldsymbol{D}^Z = \{\boldsymbol{t}_1, ..., \boldsymbol{t}_Z\}$.

Then, we have the expectation of $\mathcal{L}_{\text{dict}}$

$$\mathbb{E}[\mathcal{L}_{\text{dict}}] = \mathbb{E}_{\boldsymbol{X}} \left[ -\log \frac{\exp\langle \boldsymbol{h}, \boldsymbol{t}_+ \rangle}{\sum_{z=1}^{Z} \exp\langle \boldsymbol{h}, \boldsymbol{t}_z \rangle} \right]. \tag{5}$$

Minimizing $\mathcal{L}_{\text{dict}}$ can be regarded as minimizing a categorical cross-entropy loss of identifying the positive label embedding vector correctly, given the local feature $\boldsymbol{h}$. For examples, in Eq. 3, $\langle \boldsymbol{h}, \boldsymbol{t}_z \rangle$ serves as the confidence of classifying $\boldsymbol{h}$ as the label $y_z$. Thus, we can define the optimal probability $P^{\text{pos}}(\boldsymbol{t}_z | \boldsymbol{X})$ representing the true probability of $\boldsymbol{t}_z$ being the positive label embedding vector. Because

the label of $\boldsymbol{h}$ equals to the label of $\boldsymbol{t}_+$ by definition, it can be said that the positive label embedding vectors are sampled from the true distribution $p(\boldsymbol{t}|\boldsymbol{h})$, and the negative label embedding vectors are from the true distribution $p(\boldsymbol{t})$. Accordingly, we obtain the optimal probability as follows:

$$P^{\mathrm{pos}}(\boldsymbol{t}_z|\boldsymbol{X}) = \frac{p(\boldsymbol{t}_z|\boldsymbol{h})\Pi_{s\neq z}p(\boldsymbol{t}_s)}{\sum_{j=1}^{Z}p(\boldsymbol{t}_j|\boldsymbol{h})\Pi_{s\neq j}p(\boldsymbol{t}_s)} = \frac{\frac{p(\boldsymbol{t}_z|\boldsymbol{h})}{p(\boldsymbol{t}_z)}}{\sum_{j=1}^{Z}\frac{p(\boldsymbol{t}_j|\boldsymbol{h})}{p(\boldsymbol{t}_j)}}. \tag{6}$$

Then, we derive $\mathcal{L}_{\mathrm{dict}}^{\mathrm{optimal}}$ by using $P^{\mathrm{pos}}(\boldsymbol{t}_+|\boldsymbol{X})$ as the optimal value for $\frac{\exp\langle\boldsymbol{h},\boldsymbol{t}_+\rangle}{\sum_{j=1}^{Z}\exp\langle\boldsymbol{h},\boldsymbol{t}_j\rangle}$ in Eq. (5). Thus, assuming uniform label distribution of $\boldsymbol{h}$, we obtain the inequality

$$\mathbb{E}\left[\mathcal{L}_{\mathrm{dict}}\right] \geq \mathbb{E}\left[\mathcal{L}_{\mathrm{dict}}^{\mathrm{optimal}}\right] = \mathbb{E}_{\boldsymbol{X}}\left[-\log\frac{\frac{p(\boldsymbol{t}_+|\boldsymbol{h})}{p(\boldsymbol{t}_+)}}{\sum_{j=1}^{Z}\frac{p(\boldsymbol{t}_j|\boldsymbol{h})}{p(\boldsymbol{t}_j)}}\right] \tag{7}$$

$$= \mathbb{E}_{\boldsymbol{X}}\left[-\log\frac{\frac{p(\boldsymbol{t}_+|\boldsymbol{h})}{p(\boldsymbol{t}_+)}}{\frac{p(\boldsymbol{t}_+|\boldsymbol{h})}{p(\boldsymbol{t}_+)} + \sum_{\boldsymbol{t}_j\in\boldsymbol{D}_{\mathrm{neg}}}\frac{p(\boldsymbol{t}_j|\boldsymbol{h})}{p(\boldsymbol{t}_j)}}\right] \tag{8}$$

$$= \mathbb{E}_{\boldsymbol{X}}\left\{\log\left[1 + \frac{p(\boldsymbol{t}_+)}{p(\boldsymbol{t}_+\mid\boldsymbol{h})}\sum_{\boldsymbol{t}_j\in\boldsymbol{D}_{\mathrm{neg}}}\frac{p(\boldsymbol{t}_j\mid\boldsymbol{h})}{p(\boldsymbol{t}_j)}\right]\right\} \tag{9}$$

$$\approx \mathbb{E}_{\boldsymbol{X}}\left\{\log\left[1 + \frac{p(\boldsymbol{t}_+)}{p(\boldsymbol{t}_+\mid\boldsymbol{h})}(Z-1)\mathbb{E}_{\boldsymbol{t}_j\sim p(\boldsymbol{t}_j)}\frac{p(\boldsymbol{t}_j\mid\boldsymbol{h})}{p(\boldsymbol{t}_j)}\right]\right\} \tag{10}$$

$$= \mathbb{E}_{\boldsymbol{X},\boldsymbol{t}_+}\left\{\log\left[1 + \frac{p(\boldsymbol{t}_+)}{p(\boldsymbol{t}_+\mid\boldsymbol{h})}(Z-1)\right]\right\} \tag{11}$$

$$\geq \mathbb{E}_{\boldsymbol{X},\boldsymbol{t}_+}\left\{\log\left[\frac{p(\boldsymbol{t}_+)}{p(\boldsymbol{t}_+\mid\boldsymbol{h})}(Z-1)\right]\right\} \tag{12}$$

$$= -I(\boldsymbol{h},\boldsymbol{t}_+) + \log(Z-1) \geq -I(\boldsymbol{h},y) + \log(Z-1). \tag{13}$$

Eq. (10) is derived from Oord et al. (2018), according to which the approximation becomes more accurate as $Z$ increases. By the definition of mutual information, we have $I(\boldsymbol{h},\boldsymbol{t}_+) = \mathbb{E}_{\boldsymbol{X},\boldsymbol{t}_+}\log\frac{p(\boldsymbol{t}_+|\boldsymbol{h})}{p(\boldsymbol{t}_+)}$. Due to the data processing inequality (Shwartz-Ziv & Tishby, 2017), we have $I(\boldsymbol{h},y) \geq I(\boldsymbol{h},\boldsymbol{t}_+)$, and thereby Eq. (13). Given the derived inequality $\mathbb{E}[\mathcal{L}_{\mathrm{dict}}] \geq \log(Z-1) - I(\boldsymbol{h},y)$, minimization of $\mathcal{L}_{\mathrm{dict}}$ maximizes a lower bound of $I(\boldsymbol{h},y)$.

## C    EXPERIMENTS WITH LAYER-WISE PREDICTION

Table I: Layer-wise test errors for FC and Conv models on CIFAR-10. Layer 3 correspond to the final classifier layer.

| | Layer 1 | Layer 2 | Layer 3 |
|---|---|---|---|
| FC | $70.95 \pm 0.25\%$ | $\mathbf{34.22} \pm 0.24\%$ | $\underline{34.62} \pm 0.36\%$ |
| Conv | $48.14 \pm 0.22\%$ | $\mathbf{25.42} \pm 0.25\%$ | $\underline{25.63} \pm 0.31\%$ |

Table II: Layer-wise test errors for FC and Conv models on CIFAR-100. We report average test errors across 5 runs.

| | Layer 1 | Layer 2 | Layer 3 |
|---|---|---|---|
| FC | $72.82 \pm 0.31\%$ | $\underline{66.36} \pm 0.15\%$ | $\mathbf{65.85} \pm 0.19\%$ |
| Conv | $71.11 \pm 0.21\%$ | $\underline{50.37} \pm 0.28\%$ | $\mathbf{49.78} \pm 0.30\%$ |

This section evaluates the performance of layer-wise predictions in models trained with $\mathcal{L}_{\mathrm{dict}}$ and adaptive label embeddings. Initially, we scrutinize the individual layer-wise predictions for the FC, Conv, and VGG8B architectures. Given the shallow nature of FC and Conv architectures, we further assess the VGG8B architecture using a weighted sum of layer-wise predictions as the global prediction.

### C.1    METHOD

Models trained with $\mathcal{L}_{\mathrm{dict}}$ can generate layer-wise predictions by leveraging Eq. (3), namely

$$\hat{y} = \arg\max_z\langle\boldsymbol{h},\boldsymbol{t}_z'\rangle, \ \boldsymbol{t}' \in \{\mathrm{pool}_l(\boldsymbol{t}_z)|\boldsymbol{t}_z \in \boldsymbol{D}^Z\}. \tag{14}$$

Suppose that we have a model consisting of $L$ layers $\{f_1, ..., f_L\}$, where $f_L$ is the final linear classifier layer. Then, for $l < L$, we can obtain a layer-wise embedding vector $\boldsymbol{t}' \in \{\text{pool}_l(\boldsymbol{t}_z)|\boldsymbol{t}_z \in \boldsymbol{D}^Z\}$. By leveraging the similarity $\langle \boldsymbol{h}, \boldsymbol{t}'_z \rangle$ as the confidence of classifying $\boldsymbol{h}$ as the label $y_i$, we obtain layer-wise prediction by choosing the label with the highest confidence.

## C.2 SINGLE PREDICTION

Since both FC and Conv architectures consist of three layers, we present their performance together.

**Results for the FC and Conv architectures.** Table I and Table II summarize the performance of layer-wise predictions. On CIFAR-10, Layer 2 predictions outperform the predictions from the final classifier layer for both FC and Conv models.

**Results for the VGG8B architecture.** Figure II displays the prediction performance for all layers of VGG8B. In both datasets, while the final layers provide the most accurate predictions, those after Layer 4 are only slightly less accurate than the final layers.

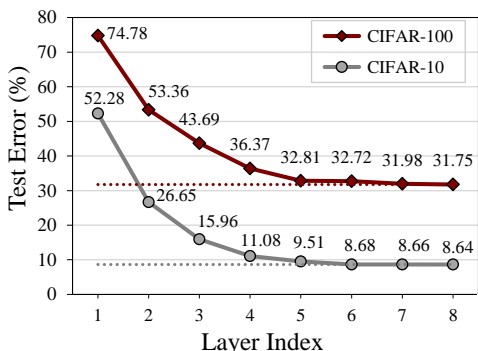

Figure II: Test errors across layers of the VGG8B models. The 8-th layer corresponds to the final linear classifier layer.

## C.3 WEIGHTED SUM OF LAYER-WISE PREDICTION AS THE GLOBAL PREDICTION

**Weighted sum.** For each intermediate layer $l$, we can obtain a confidence vector $\hat{\boldsymbol{y}}_l = [\langle \boldsymbol{h} \ \boldsymbol{t}'_1 \rangle \ \langle \boldsymbol{h}, \boldsymbol{t}'_2 \rangle \ ... \ \langle \boldsymbol{h}, \boldsymbol{t}'_Z \rangle]^\top$. Then, we obtain the global prediction $\bar{\boldsymbol{y}}$ as the weighted sum of $\hat{\boldsymbol{y}}_l$ across $l$, such that $\bar{\boldsymbol{y}} = \sum_{l=1}^{L-1} \lambda_l \hat{\boldsymbol{y}}_l$, where $\lambda_l$ is the coefficient controlling the influence of $\hat{\boldsymbol{y}}_l$ on $\bar{\boldsymbol{y}}$. We evaluate $\bar{\boldsymbol{y}}$ by setting $\lambda_l = l^p$, such that $0 < p <= 10$. The trained VGG8B models from Section 5.2 are employed for these global predictions.

**Results.** Figure III details the global prediction results for different $p$ values. When $p = 1$, all layers contribute equally, leading to a performance drop compared to the final classifier layer. As $p$ increases, deeper layers exert greater influence on $\bar{\boldsymbol{y}}$. Higher $p$ values generally enhance predictions above the baseline, notably on CIFAR-100. Yet, an overwhelming influence from deeper layers can degrade performance, regressing toward the baseline.

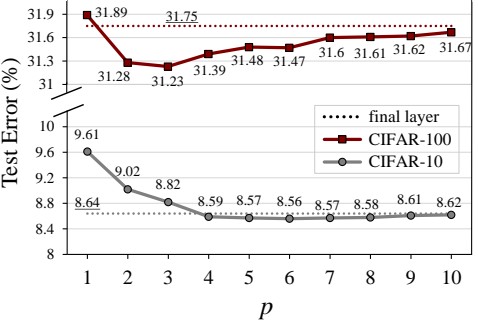

Figure III: Test errors for global prediction at different $p$ values. Here, $p$ adjusts the confidence level of each layer, such that $\lambda_l = l^p$. Baselines are derived from the average test errors of the final classifier layer. The underlined values correspond to the baselines.

# D MULTIPLE LABEL EMBEDDINGS PER LABEL

This section delves into a situation where each class has multiple label embeddings. Specifically, we examine an ideal scenario where the variation within each class label is accurately and specifically identified. CIFAR-100, composed of 20 super-labels with 5 sub-labels each, serves as our testbed. Training on CIFAR-100 and performing inference on CIFAR-20 essentially equates to employing 5 unique label embeddings for each super-label. This approach enables us to leverage the exact knowledge of 5 distinct variations for each label.

**Setup.** We employ two distinct methods for inference to leverage the hierarchical relationship between CIFAR-100 and CIFAR-20 labels. The `Mean` method averages the sub-label embeddings

within each super-label, creating a unified super-label embedding—for example, averaging embeddings of "man," "woman," "boy," "girl," and "baby" to represent "people." During inference, the model predicts the super-label by identifying which averaged embedding is most similar to the local features from the last layer $\bar{h}_{L-1}$, utilizing a similarity metric as outlined in Eq. (3). On the other hand, `Super` method predicts the super-label based on the closest proximity of $\bar{h}_{L-1}$ to any sub-label embedding within a super-label.

For comparative analysis, we introduce two baseline approaches that utilize conventional feedforward processes for inference. `Naive` approach uses the super-label of the predicted sub-label as the final prediction. We utilize sub-label predictions by the models trained on CIFAR-100 from Section 5.2. For instance, if the model classifies an image as "baby," the associated super-label "people" is used as the final prediction. Meanwhile, `E2E` represents an end-to-end training method, where models are trained using $\mathcal{L}_{\text{dict}}$ on CIFAR-20 super-labels.

**Results and analysis.**   Table III indicates that `Super` achieves the best performance overall. However, `Mean` does not exhibit a significant advantage over `Naive`, even showing a performance decline for the VGG8B model. This could be attributed to the presence of outlier sub-labels in CIFAR-100, which can negatively skew the computed mean. For instance, as depicted in Figure VII, the "forest" sub-label embedding is closer to the "trees" embedding than to other sub-labels in the "large natural outdoor scenes" category. Consequently, an averaged embedding for "large natural outdoor scenes" may not effectively represent the nuances of this super-label.

Table III: Comparison of test errors on CIFAR-20.

|  | FC | Conv | VGG |
|---|---|---|---|
| E2E | **56.65** | 40.72 | 23.42 |
| Naive | 65.85 | 49.78 | 31.75 |
| Super | 60.95 | **37.56** | **21.84** |
| Mean | 60.98 | 48.61 | 32.26 |

It is also important to note that the semantic relationships learned by $\mathcal{L}_{\text{dict}}$ do not always align with the dataset's predefined hierarchy. For example, CIFAR-100 categorizes "chimpanzee" under "large omnivores and herbivores" alongside "elephant, cattle, camel, and kangaroo." Yet, semantically, "chimpanzee" aligns more with "people," as reflected in the learned label embeddings in Figure VII. This observation highlights that models trained with $\mathcal{L}_{\text{dict}}$ intuitively grasp the semantic relationships between labels, while learning a less inherently semantic structure, such as a taxonomy, requires explicit supervision.

## E   Feature segments $K$

Figure IV illustrates how performance varies across different $K$ values for CIFAR-10 and CIFAR-100. In our FC layers, we divide a local feature $h_{\text{flat}} \in \mathbb{R}^{CK}$ into $K$ segments of $C$-dimensional feature vectors as mentioned in Section 4.2. For our FC models, $CK = 3072$ across all intermediate layers. This transformation effectively converts the flat feature into a grid-like feature $h \in \mathbb{R}^{C \times K}$. Then, $\mathcal{L}_{\text{dict}}$ optimizes the average similarity between label embedding vectors and feature vectors across $K$ segments. While the optimal number for $K$ may vary across datasets, it is evident that the segmentation of features consistently leads to improved performance compared to the baseline $K = 1$.

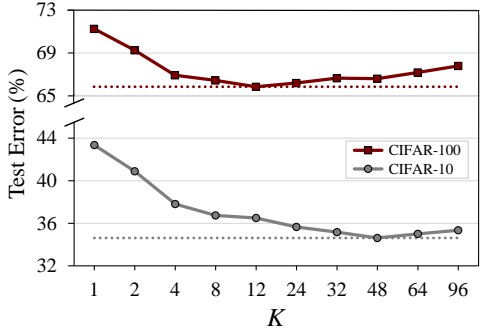

Figure IV: Performance of FC models with varying numbers of feature segments $K$.

## F   DCL variants for Parallel Training

**Setups.**   `DCL`, our default approach, updates label embeddings following every intermediate layer's forward pass. While our experiments exclusively employ sequential training, we also explore parallel training scenarios where each layer operates on a distinct GPU in parallel. With `DCL`, label embeddings can be updated via layer-wise gradients averaged across all intermediate layers. However, this averaging, which integrates concurrent error signals from all layers, might negatively impact the weight updates. Thus, we propose two versions better suited for parallel training. `DCL-O` updates

label embeddings exclusively using the error signals from the last intermediate layer. In contrast, `DCL-LD` employs layer-wise dictionaries $\boldsymbol{D}_l^Z = \{\boldsymbol{t}_z \mid \boldsymbol{t}_z \in \mathbb{R}^{C_D^l}, \ z \in \{1, ..., Z\}\}$, allowing for parallel updates of layer-wise label embeddings.

Table IV: Comparison of `DCL` variants trained on the VGG8B architecture. *Memory* denotes the peak GPU memory consumption measured during single GPU training with a batch size of 128. $\Delta\theta$ represents the increase in the number of parameters compared to the BP baseline.

| Method | MNIST | | | CIAFR-10 | | | CIAFR-100 | | | STL-10 | | |
|---|---|---|---|---|---|---|---|---|---|---|---|---|
| | $\Delta\theta$ | *Err.* | *Memory* | $\Delta\theta$ | *Err.* | *Memory* | $\Delta\theta$ | *Err.* | *Memory* | $\Delta\theta$ | *Err.* | *Memory* |
| `DCL` | **5.12K** | 0.33 | **580 MiB** | **5.12K** | 8.64 | **747 MiB** | **51.2K** | 31.75 | **751 MiB** | **5.12K** | 22.87 | **1589 MiB** |
| `DCL-O` | **5.12K** | **0.32** | **580 MiB** | **5.12K** | 8.68 | **747 MiB** | **51.2K** | 34.58 | **751 MiB** | **5.12K** | 23.56 | **1589 MiB** |
| `DCL-LD` | 21.8K | 0.34 | 581 MiB | 21.8K | **8.45** | 749 MiB | 218K | **31.64** | 766 MiB | 21.8K | **22.59** | 1593 MiB |

**Results.** Table IV compares performance and parameter/memory costs for `DCL`, `DCL-O`, and `DCL-LD` across four datasets: MNIST, CIAFR-10, and CIAFR-100, and STL-10. `DCL-LD`'s layer-wise dictionaries might provide advantages in certain contexts, as seen in its competitive error rates for CIAFR-100 and STL-10, albeit with higher parameter/memory cost. `DCL` and `DCL-O` maintain a balance between efficiency and performance, with `DCL-O` slightly edging out in error rate reduction in less complex datasets. The increased parameter count for `DCL-LD` suggests a trade-off between representational capacity and resource efficiency, which may be justified by its performance in more complex tasks.

## G   EXPERIMENTS WITH OTHER ARCHITECTURES.

Table V: Comparison between $\mathcal{L}_{\text{dict}}$ and $\mathcal{L}_{\text{contrast}}$ across various architectures on CIFAR-10.

| | ResNet-32 | | | ResNet-32-W | | | ViT | | | MLP-Mixer | | |
|---|---|---|---|---|---|---|---|---|---|---|---|---|
| | $\Delta\theta$ | *Err.* | *Memory* | $\Delta\theta$ | *Err.* | *Memory* | $\Delta\theta$ | *Err.* | *Memory* | $\Delta\theta$ | *Err.* | *Memory* |
| BP | **0** | **6.701** | 3179 MiB | **0** | **4.781** | 13433MiB | **0** | **16.25** | 5300 MiB | **0** | **17.23** | 6361 MiB |
| $\mathcal{L}_{\text{contrast}}$ | 73.6K | 32.71 | 1617 MiB | 266K | 20.02 | 6378 MiB | 394K | 32.95 | 1354 MiB | 394K | 31.67 | 1468 MiB |
| $\mathcal{L}_{\text{dict}}$ | 0.64K | 25.19 | **1617 MiB** | 2.56K | 14.36 | **5350 MiB** | 5.12K | 32.63 | **1348 MiB** | 5.12K | 23.51 | **1445 MiB** |

We evaluate $\mathcal{L}_{\text{dict}}$ against $\mathcal{L}_{\text{contrast}}$ on CIFAR-10 using a range of architectures: ResNet-32 (He et al., 2016), its wider-channel variant ResNet-32-W, ViT (Dosovitskiy et al., 2020), and MLP-Mixer (Tolstikhin et al., 2021). These architectures consist of multiple modules, each embodying a functional unit and containing multiple fully connected or convolutional layers. For instance, a single module of MLP-Mixer is a combination of channel-mixing and token-mixing layers, so that each module effectively blends information both channel-wise and token-wise. Our experimental setup applies $\mathcal{L}_{\text{dict}}$ at a module level, incorporating module-wise BP. However, unlike in $\mathcal{L}_{\text{contrast}}$, we do not employ module-wise auxiliary networks. Please refer to Appendix J.2.3 for additional information.

**Results.** Table V demonstrates that $\mathcal{L}_{\text{dict}}$ outperforms $\mathcal{L}_{\text{contrast}}$ across various architectures while requiring fewer parameters and less memory. Although the performance difference for ViT is marginal, a considerable gap is observed in ResNet-32, ResNet-32-W, and MLP-Mixer.

## H   THE CHOICE OF SIMILARITY MEASURE

In Figure V, we compare dot product and cosine similarity across different temperature parameter $\tau$. Although cosine similarity is a commonly employed similarity measure in contrastive learning, it consistently delivers inferior performance when compared to dot product, irrespective of $\tau$.

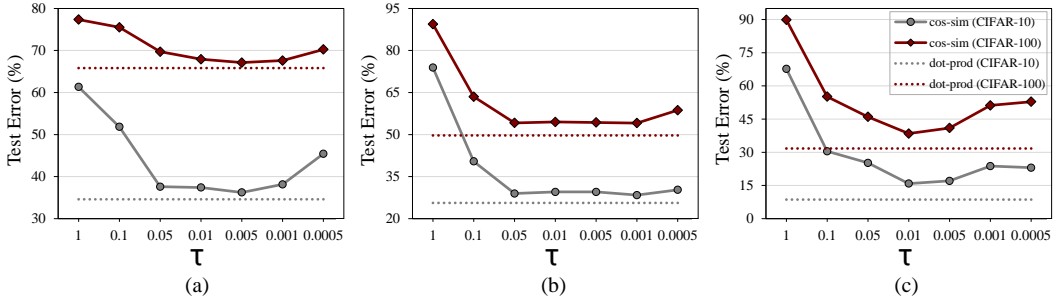

Figure V: Cosine similarity vs. dot product across different architectures. (a) FC, (b) Conv, and (c) VGG8B. $\tau$ denotes the temperature hyperparameter for cosine similarity. Across all tested architectures and datasets, dot product consistently outperforms cosine similarity.

## I IN RELATION TO NEURAL COLLAPSE

**Neural Collapse.** During the late stages of training, various phenomena collectively known as neural collapse appear at the layer preceding the final classifier. The most relevant to our study is the simplification to nearest class center (NCC): the network classifier tends to select the class whose train class mean is nearest (Papyan et al., 2020). Previous research have identified the gradual emergence of neural collapse at local layers (Ben-Shaul & Dekel, 2022; Ben-Shaul et al., 2023; Rangamani et al., 2023). Moreover, it has been shown that inducing NCC in these layers through specialized local losses can be advantageous (Ben-Shaul & Dekel, 2022; Elsayed et al., 2018).

**Regarding our work.** By minimizing $\mathbb{E}[\mathcal{L}_{\text{dict}}(\boldsymbol{h}, \boldsymbol{D}_Z)]$, our approach can be seen as encouraging NCC at local layers by aligning the local class center $\frac{1}{N_z} \sum_{n=1}^{N_z} \boldsymbol{h}_n^z$ with label embedding vectors $\boldsymbol{t}_z$, akin to the motivation of using local losses by Ben-Shaul & Dekel (2022); Elsayed et al. (2018). However, Ben-Shaul & Dekel (2022); Elsayed et al. (2018) focus on local features within an end-to-end BP framework, whereas our work investigates locally decoupled layers, treating each as the final layer before the classifier. Our results indicate that global feedback, typical of BP, may not be crucial for neural collapse. In Figure II, we observe the improvement of layer-wise prediction through layers, consistent with Ben-Shaul et al. (2023). Moreover, in Appendix D, clustering by semantic similarity, as opposed to predefined taxonomy, mirrors the clustering to semantic labels observed by encouraging NCC in self-supervised learning (Ben-Shaul et al., 2023).

## J IMPLEMENTATION DETAILS

### J.1 DATASETS

**FC and Conv experiments.** For the Conv and FC architectures, we test our method on MNIST (LeCun, 1998), CIFAR-10, and CIFAR-100 (Krizhevsky et al., 2009) datasets, in Table 2 and Table 1. The MNIST dataset consists of 60000 training and 10000 test samples, with 10 label classes. Each sample is a $28 \times 28$ grayscale image. CIFAR-10 and CIFAR-100 provide 50000 training and 10000 testing RGB images of size $32 \times 32$, but with 10 and 100 label classes respectively. All three datasets maintain uniform label distributions across training and test sets.

**VGG8B experiments.** In Table 3, employing the VGG8B (Simonyan & Zisserman, 2015; Nøkland & Eidnes, 2019) architecture, our method was evaluated on datasets including MNIST, Fashion-MNIST (Xiao et al., 2017), CIFAR-10, CIFAR-100, SVHN (Netzer et al., 2011), and STL-10 (Coates et al., 2011) datasets. Fashion-MNIST (F-MNIST) has 60,000 training and 10,000 test grayscale images with 10 label classes. SVHN, with $32 \times 32$ RGB images, comprises 73257 training, 531131 extra training, and 26032 test images. We adopt both training sets, following Nøkland & Eidnes (2019). STL-10 provides 5,000 training and 8,000 test RGB images ($96 \times 96$). Labels of Fashion-MNIST, CIFAR-10, CIFAR-100, and STL-10 are uniform distributions for training and test samples. Nøkland & Eidnes (2019) also tested their models on Kuzushiji-MNIST (Clanuwat et al., 2018).

Because the newer, available version of Kuzushiji-MNIST is different, we exclude Kuzushiji-MNIST from the experiment.

## J.2 TRAINING DETAILS

Table VI: FC and Conv architectures. `DCL` and BP share the same architectures, except for $D^Z$. Likewise, `DCL`, `FF` (Hinton, 2022), and `SymBa` (Lee & Song, 2023) share the same FC architectures. $K$ refers to the feature segments explained in Section E.

|  | FC (MNIST) | FC (CIFAR-10) | FC (CIFAR-100) | Conv | Conv (`DCL-S` CIFAR-100) |
|---|---|---|---|---|---|
| $D^Z$ | $10 \times 98$ | $10 \times 64$ | $100 \times 256$ | $10 \times 256$ ($100 \times 256$) | $100 \times 512$ |
| $K$ | 8 | 48 | 12 |  |  |
| Input Size | 784 | 3072 | 3072 | 3x32x32 | 3x32x32 |
| Unit 1 | fc 1024 | fc 3072 | fc 3072 | conv($3 \times 3 \times 64, 1, 1$) | conv($3 \times 3 \times 64, 1, 1$) |
| Unit 2 | ReLU | ReLU | ReLU | Batchnorm | Batchnorm |
| Unit 3 | Layernorm | Layernorm | Dropout 0.3 | ReLU | ReLU |
| Unit 4 | fc 1024 | fc 3072 | LayerNorm | conv($3 \times 3 \times 256, 2, 1$) | conv($3 \times 3 \times 256, 2, 1$) |
| Unit 5 | ReLU | ReLU | fc 3072 | Batchnorm | Batchnorm |
| Unit 6 | Layernorm | Layernorm | ReLU | ReLU | ReLU |
| Unit 7 | fc 10 | fc 10 | Dropout 0.3 | Global Avg Pooling | conv($3 \times 3 \times 512, 2, 1$) |
| Unit 8 |  |  | Layernorm | fc 10 (100) | Batchnorm |
| Unit 9 |  |  | fc 100 |  | ReLU |
| Unit 10 |  |  |  |  | Global Avg Pooling |
| Unit 11 |  |  |  |  | fc 100 |

Table VII: Hyperparameters for training the FC, Conv, VGG8B architectures.

| FC | | | |
|---|---|---|---|
| Dataset | Epoch | Learning Rate | Decay Milestones |
| MNIST | 150 | 0.0005 | 50 100 125 |
| CIFAR-10 | 400 | 0.0002 | 50 150 200 350 |
| CIFAR-100 | 200 | 0.0001 | 50 100 150 |

| Conv | | | |
|---|---|---|---|
| Dataset | Epoch | Learning Rate | Decay Milestones |
| MNIST | 150 | 0.0075 | 50 75 100 125 |
| CIFAR-10 | 500 | 0.0075 | 100 200 300 400 450 |
| CIFAR-100 | 400 | 0.0075 | 100 200 250 300 350 |

| VGG8B | | | |
|---|---|---|---|
| Dataset | Epoch | Learning Rate | Decay Milestones |
| MNIST | 100 | 0.0005 | 50 75 89 94 |
| F-MNIST | 200 | 0.0005 | 100 150 175 188 |
| CIFAR-10 | 400 | 0.0005 | 200 300 350 375 |
| CIFAR-100 | 400 | 0.0005 | 200 300 350 375 |
| SVHN | 100 | 0.0003 | 50 75 89 94 |
| STL-10 | 400 | 0.0005 | 200 300 350 375 |

Our experiments were conducted using Pytorch. Across all architectures, we use the AdamW optimizer (Loshchilov & Hutter, 2018) with the default Pytorch setting: $\beta_1 = 0.9$, $\beta_2 = 0.999$, and weight_deacy $= 0.01$. Table VII details the training hyperparameters for every architecture.

Table VIII: Backbone architecture of VGG8B. All convolution layers utilize $3 \times 3$ kernels, with a stride and zero-padding set to 1.

| | Layer 1 | Layer 2 | Layer 3 | Layer 4 | Layer 5 | Layer 6 |
|---|---|---|---|---|---|---|
| Unit 1 | conv 128 | conv 256 | conv 256 | conv 512 | conv 512 | conv 512 |
| Unit 2 | Batchnorm | Batchnorm | Batchnorm | Batchnorm | Batchnorm | Batchnorm |
| Unit 3 | ReLU | ReLU | ReLU | ReLU | ReLU | ReLU |
| Unit 4 | Dropout | Dropout | Dropout | Dropout | Dropout | Dropout |
| Unit 5 | | Maxpool $2 \times 2$ | | Maxpool $2 \times 2$ | Maxpool $2 \times 2$ | Maxpool $2 \times 2$ |

| Dropout Rates across Different Datasets | | | | | |
|---|---|---|---|---|---|
| | MNIST | F-MNIST | CIFAR-10 | CIFAR-100 | SVHN | STL-10 |
| Dropout rate | 0.1 | 0.1 | 0.05 | 0.05 | 0.05 | 0.1 |

Table IX: Fully connected layers succeeding the VGG8B backbone. Feature maps from the final convolutional layers are transformed into feature vectors. The initial dimension of the "fc" layers represents the feature vectors' dimension. Dropout rates are identical to those used in the backbone.

| | MNIST, F-MNIST | | CIFIAR-10 (100), SVHN | | STL-10 | |
|---|---|---|---|---|---|---|
| | Layer 7 | Layer 8 | Layer 7 | Layer 8 | Layer 7 | Layer 8 |
| Unit 1 | fc 512×1024 | fc 1024×10 | fc 2048×1024 | fc 1024×10 (100) | fc 4608×1024 | fc 1024×10 |
| Unit 2 | Batchnorm | | Batchnorm | | Batchnorm | |
| Unit 3 | Dropout | | Dropout | | Dropout | |

### J.2.1 FC AND CONV EXPERIMENTS.

For Table 1 and 2, our models and their BP baselines are trained using the same training hyperparameters. The architecture details are listed in Table VI. All FC and Conv experiments employ a learning rate decay rate of $0.5$ and batch size of $512$.

### J.2.2 VGG8B EXPERIMENTS.

For Table 3, VGG8B architectures come in three variants, each tailored to a specific input image size, but they all share the backbone layers listed in Table VIII. Table IX summarizes the variations in the fully connected heads. For the label embedding dictionary $D^Z \in \mathbb{R}^{Z \times C_D}$, where $Z$ equals the number of label classes, we use $C_D = 512$ across all datasets. For the "Layer 7" in Table IX, we segment the output 1024-dimensional vector into 2 segments of 512-dimensional vectors for $\mathcal{L}_{\text{dict}}$ in Eq. (2). A batch size of 128 is used for all experiments.

### J.2.3 EXPERIMENTS WITH RESNET, VIT, AND MLPMIXER.

**ResNet-32 and ResNet-32-W.** ResNet-32 is structured with 15 residual blocks, each containing 2 convolutional layers. These blocks are treated as individual modules. ResNet-32-W is structurally identical to ResNet-32, with the key distinction being that its channel dimension is four times larger than that of ResNet-32. Training details of ResNet-32 and ResNet-32-W are in Appendix J.2.6.

**ViT.** We employ a vision transformer (ViT) containing 6 transformer encoder modules. Each module includes multi-head attention (MHA) and a multilayer perceptron (MLP), with both MHA and MLP having two fully connected (FC) layers. ViT's configuration involves 8 attention heads, $4 \times 4$ image patch size, 512-dimensional tokens, and a single class token.

In line with the original design of the class token in ViT, we optimize the similarity between label embedding vectors $\boldsymbol{t}_z$ and the class token vector. We use the same CIFAR-10 training hyperparameters utilized for VGG8B in Table VII.

**MLP-Mixer.** In our experiments, MLP-Mixer employs 6 modules, each featuring a channel-mixing and a token-mixing layer. With a 4x4 image patch size on CIFAR-10, each local module processes 64 tokens of 512-dimensional vectors. Training hyperparameters for CIFAR-10 are identical to those used for VGG8B, as shown in VII.

### J.2.4 IMPLEMENTING $\mathcal{L}_{\mathrm{contrast}}$ AND $\mathcal{L}_{\mathrm{feat}}$.

**Training with $\mathcal{L}_{\mathrm{contrast}}$.** We train VGG8B and ResNet-32 models with auxiliary networks for Figure 1, Figure 3. Table 3, and Appendix A because $\mathcal{L}_{\mathrm{contrast}}$ entails the use of auxiliary networks.

For VGG8B, we used the same auxiliary network architecture and training hyperparameters used by Nøkland & Eidnes (2019). The model incorporates a layer-wise fully connected auxiliary network, transforming $A$-dimensional vectors into 128-dimensional ones. $A$ equals 2048 except for MNIST and FashionMNIST which use $A = 1024$. The local feature maps $\boldsymbol{h}$ undergo adaptive average-pooling, ensuring the flattened local outputs are standardized to 1024-dimensional vectors. Training hyperparameters are available in Appendix J.2.2.

For ResNet-32, we adopt the same auxiliary network architecture as used by Wang et al. (2020), comprising three convolutional layers and two fully connected layers. The training settings are detailed in Appendix J.2.6.

**Training with LL-contrec.** For LL-contrec in Section 5.2, we combine $\mathcal{L}_{\mathrm{contrast}}$ with an image reconstruction loss $\mathcal{L}_{\mathrm{rec}}$ used in estimating $I(\boldsymbol{h}, \boldsymbol{x})$ (see Appendix J.2.7 for more details). Then, we have $\mathcal{L}_{\mathrm{contrec}} = \lambda_1 \mathcal{L}_{\mathrm{contrast}} + \lambda_2 \mathcal{L}_{\mathrm{rec}}$, where $\lambda_1 = \alpha\beta$ and $\lambda_2 = \alpha(1 - \beta)$ (Wang et al., 2020). $\beta$ progressively increases as layers $l$ deepen, such that $\beta = \frac{l}{L-1}$, $l \in \{1, ..., L - 1\}$.

**Training with $\mathcal{L}_{\mathrm{feat}}$.** For Figures 3, Table 1, Table 2, and I, we train the models using $\mathcal{L}_{\mathrm{feat}}$, with the training hyperparameters designated for each architecture. We can derive $\mathcal{L}_{\mathrm{feat}}$ by using $f_\phi(\boldsymbol{h}) = \boldsymbol{h}$ in Eq. (1). For the fully connected layers, we directly use the output vector as $\boldsymbol{h}$. For CNN, flattening a feature map $\boldsymbol{h_{map}} \in \mathbb{R}^{C \times H \times W}$ leads to an unwieldy vector size. To address this, we employ the average local feature $\boldsymbol{h} := \frac{1}{K} \sum_{k=1}^{K} \boldsymbol{h}^k$, where $K = HW$.

### J.2.5 REPLICATED FINDINGS.

For benchmarks not available in published papers, we reproduce baseline models with their open-source codes.

**For FC and Conv architectures.** We train FF (Hinton, 2022), PFF (Ororbia & Mali, 2023), and DRTP (Frenkel et al., 2021) on CIFAR-100 for Table 1 and Table 2. We apply the same hyperparameters and architectures prescribed for CIFAR-10. To modify the architectures for CIFAR-100, we upscale the output dimension of the final classifier from 10 to 100. This explains the increase in the number of parameters in Table 1 and Table 2. Likewise, for PFF, which originally takes $28 \times 28$ grayscale images, we upscale the input dimension to accommodate $32 \times 32$ RGB images.

**LL-bpf experiments.** Because LL-bpf is the BP-free version of LL-predsim, we use the same training hyperparameters designated for LL-predsim to train LL-bpf models for Table 3. LL-bpf models utilize two separate layer-wise projections. The first projection maps one-hot encoded labels to 128-dimensional vectors using fixed random weights to prevent BP. The second projection requires adaptive pooling, such that flatten local outputs become 4096-dimensional vectors. Then, a fully connected layer projects the 4096-dimensional vectors to 128-dimensional vectors. Layers receive error signals from the binary cross entropy loss between the projected 128-dimensional vectors. The error signals are propagated by feedback alignment (Lillicrap et al., 2016) to prevent BP.

### J.2.6 TRAINING SETTINGS FOR RESNET-32 EXPERIMENTS

We train ResNet-32 (He et al., 2016) for five separate experiments: 1) Figure 1, 2) Figure 3, and 3) Table V. The training configurations remain the same across these tasks, based on the specifications from Wang et al. (2020). Namely, using an SGD optimizer with a Nesterov momentum (Sutskever et al., 2013) of 0.9, the ResNet-32 models are trained for 160 epochs, with the L2 weight decay ratio of 1e-4, batch size of 128, and the cosine annealing with an initial learning rate of 0.8. In all our experiments, we compute local losses based on the output from each residual block. ResNet-32 comprises 16 residual blocks, resulting in 16 local losses.

### J.2.7 DETAILS ON IMPLEMENTING MUTUAL INFORMATION EXPERIMENTS.

For Figure 3 and Figure I, we extract local features from the VGG8B models trained on CIFAR-10 to estimate $I(\boldsymbol{h}, \boldsymbol{x})$ and $I(\boldsymbol{h}, y)$. We follow the same architectures and hyperparameters used by Wang et al. (2020) for these experiments. Each estimation experiment employs bilinear interpolation to align the local feature map's height×width to $32 \times 32$.

**Estimating $I(\boldsymbol{h}, y)$.** We train ResNet-32 to predict $y$ using $\boldsymbol{h}$ as input. We employ the best test accuracy as an estimate of $I(\boldsymbol{h}, y)$. Appendix J.2.6 summarizes the training setup for ResNet-32. Given the varying dimensions of these layer-wise outputs, we adjust the input channel dimension to align with the channel size of the local feature maps.

**Estimating $I(\boldsymbol{h}, \boldsymbol{x})$.** We train reconstruction networks to reconstruct $\boldsymbol{x}$ using $\boldsymbol{h}$ as input, such that $\mathcal{L}_{\text{rec}} = \mathcal{L}_{\text{BCE}}(f_\psi(\boldsymbol{h}), \boldsymbol{x})$, where $\mathcal{L}_{\text{BCE}}$ is the binary cross-entropy loss, and $f_\psi$ is a reconstruction network. We employ $1 - \mathcal{L}_{\text{BCE}}$ between the output RGB image and $\boldsymbol{x}$ as an estimate of $I(\boldsymbol{h}, \boldsymbol{x})$. $f_\psi$ consists of two convolutional layers. The initial layer aligns the channel dimension of the local feature maps to 12, and the subsequent layer transforms these maps into RGB images, ending with a sigmoid activation function. The networks are trained for 10 epochs using the default Pytorch Adam optimizer (Kingma & Ba, 2015), with an learning rate set at 0.001.

## K  VISUALIZATION

### K.1  LABEL EMBEDDINGS AND CONFUSION RATES

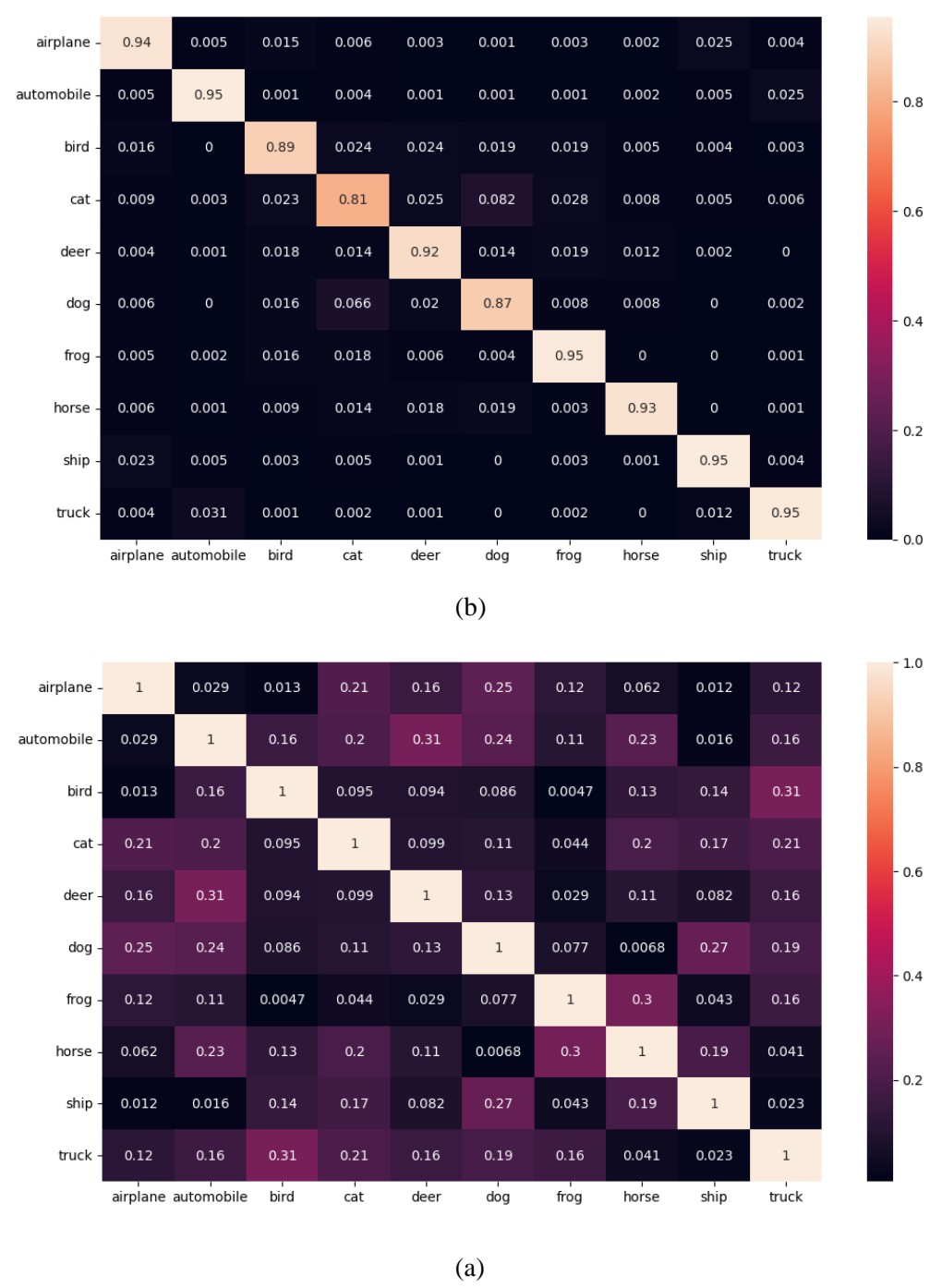

Figure VI: Label embedding similarity and confusion matrix of the VGG8B model trained using $\mathcal{L}_{\text{dict}}$ on CIFAR-10. (a) Cosine similarity matrix of label embedding vectors. Each number in a cell denotes the cosine similarity between label embedding vectors corresponding to each label pair. (b) Confusion Matrix. Labels in the x-axis are the ground truth, and labels in y-axis are predicted labels. Each cell value signifies the ratio of predicting $y_x$ as $y_y$.

## K.2 SEMANTIC RELATIONSHIP BETWEEN ADAPTIVE LABEL EMBEDDING VECTORS

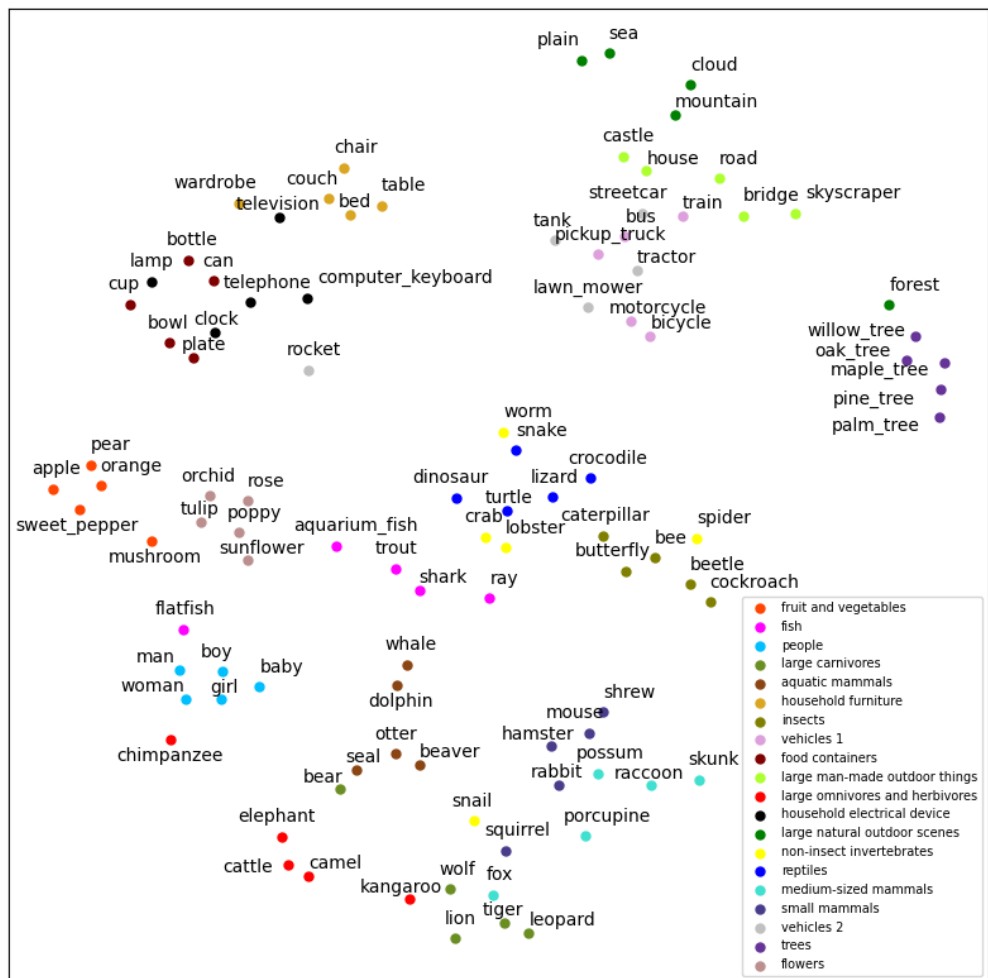

Figure VII: The t-SNE Van der Maaten & Hinton (2008) visualization of CIFAR-100 label embedding vectors. CIFAR-100 consists of 20 super-classes, each containing 5 sub-classes. The legend lists these super-classes.

## K.3    SALIENCY MAPS

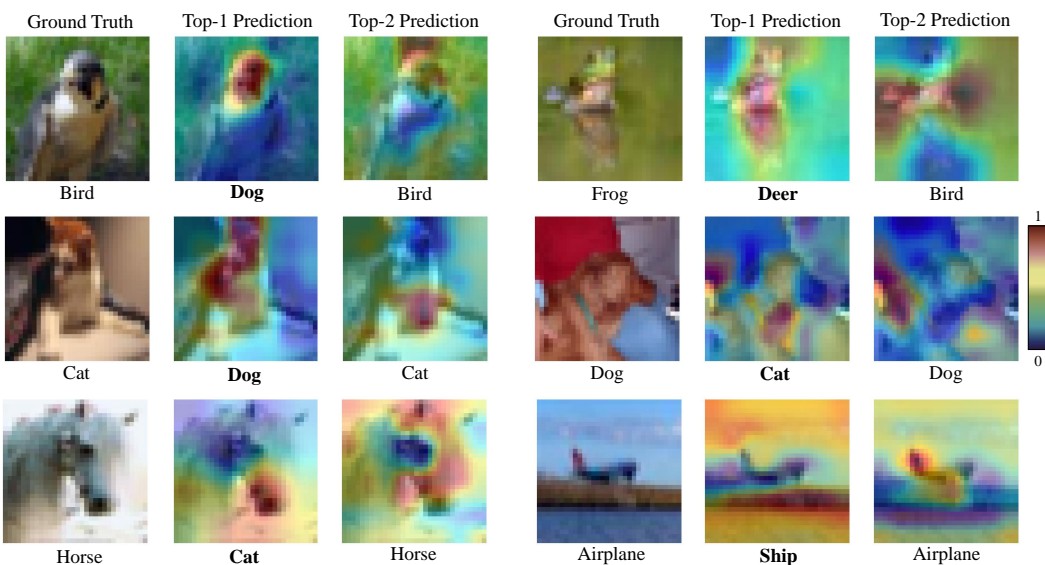

Figure VIII: Saliency maps for incorrect top-1 prediction results.

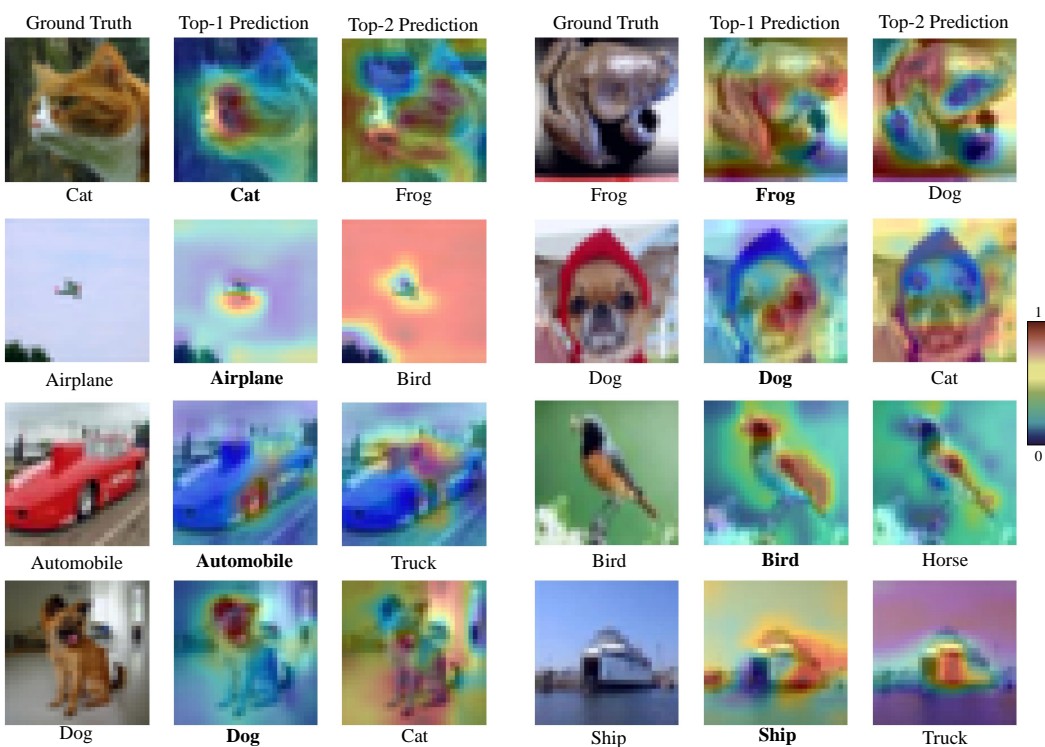

Figure IX: Saliency maps for correct top-1 prediction results.

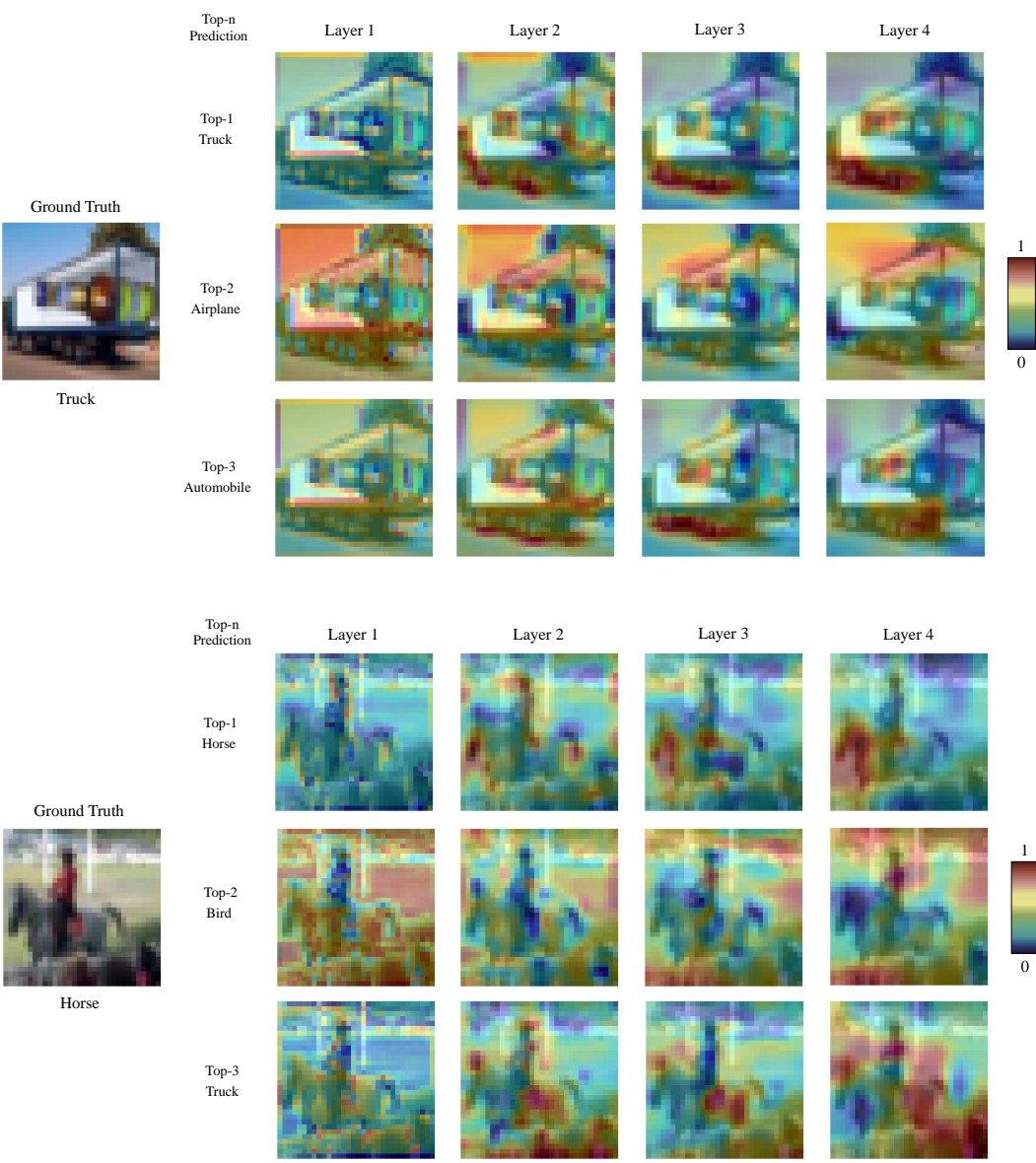

Figure X: Evolution of saliency maps across layers. The layer index denotes the source of the extracted local features.

