# OpenReview forum: "Dictionary Contrastive Learning for Efficient Local Supervision without Auxiliary Networks"
_ICLR.cc/2024/Conference — ICLR 2024 spotlight_

### Official Review · Reviewer_SF1W · 2023-10-27

**Soundness:** 3 good
**Presentation:** 3 good
**Contribution:** 3 good
**Rating:** 8
**Confidence:** 3

**Summary:**

This paper proposes a back-propagation free training method for neural networks based on a novel contrastive learning loss. This contrastive learning loss is defined in terms of a dictionary of label embeddings. The main motivation for using this loss is reduction of the mutual information between feature representations and a nuisance term, which model task-irrelevant information present in the inputs. The authors demonstrate theoretically that minimizing the proposed loss is equivalent to maximizing the mutual information of representations and labels. Furthermore, they demonstrate empirically that their algorithm reduces the mutual information of representations and the nuisance.

**Strengths:**

The authors tackle an important direction in the training of neural nets, i.e., back-propagation free training. They do so in a novel way, by proposing a new training objective based on a dictionary of label embeddings. Their contribution is also interesting through the angle of the label embeddings themselves. The resulting algorithm works well in practice, being comparable or even better than existing state-of-the-art, and while being relatively light-weight in comparison with existing approaches.

**Weaknesses:**

__W1: Presentation issues.__ Below I highlight a few presentation issues,

- The authors mention the mutual information $I$, but never formaly define it. Furthermore, even though the authors given details about how they estimate mutual information, I think an explicit pointer (e.g., Appendix D describes how we estimate the mutual information) in the text could make it the paper easier to read.

- In my view, the result the authors prove in Appendix B is part of the paper's contribution. I think authors should consider moving it to the main paper and properly make a statement about the claim, as it is used throughout the author's analysis.

__W2: Inconsistent notation.__ While in eqn. $i$ and $j$ are used to refer sample indices, in eqn. 2 the authors use $n=1,\cdots,N$. Index $i$ in eqn. 2 then means the class $i=1,\cdots,Z$ and elements of the dictionary are referred to as $\mathbf{t}\_{z} \in \mathbf{D}^{Z}$, which is confusing as well, as $z$ can be easily mistaken by $\mathbf{z} = f_{\phi}(\mathbf{h})$. Furthermore, in page 3, $y \in \mathbb{N}$ would imply an infinite amount of classes. Authors should refer to $y \in \{1,\cdots, Z\}$, as the authors do in table 4. The fact that one has a finite number of classes should be explicit.

__Minor.__ The paper contains some reference errors, especially w.r.t. arxiv references to conference papers. Here is a non-exhaustive list of wrong references,

- Yulin Wang, Zanlin Ni, Shiji Song, Le Yang, and Gao Huang. Revisiting locally supervised learning: an alternative to end-to-end training. arXiv preprint arXiv:2101.10832, 2021. (Published in ICLR 2021)
- Karen Simonyan and Andrew Zisserman. Very deep convolutional networks for large-scale image recognition. arXiv preprint arXiv:1409.1556, 2014. (Published in ICLR 2015)

__Note.__ the minor point __did not influenced my final score__.

__Post-Discussion__

The authors have corrected the presentation issues and inconsistent notation. As a result I raised my score towards 8: Accept

**Questions:**

__Q1.__ While the authors show in Appendix B that minimizing $\mathcal{L}\_{dict}$ is equivalent to maximizing a lower bound of $I(\mathbf{h}, y)$, could a similar statement be derived for $I(\mathbf{h}, r)$ or $I(r, y)$?

__Q2.__ Do the learned label embeddings have any semantic meaning? The authors could verify this through dimensionality reduction on $\mathbf{t} \in \mathbf{D}^{Z}$, in the context of a dataset with a large number of classes (i.e., CIFAR-100).

__Post-Discussion__

The authors added a visualization of label embeddings, in which  they show that the embeddings learned by their method carry semantic meaning.

---

> ### Author Response · Authors · 2023-11-20
> **Response to Reviewer SF1W**
>
> ## W1 (Mutual information)
> Many thanks for this advice and suggestion. Following Reviewer SF1W’s feedback, we have included the definition of mutual information in Section 4.1. That is, we added the definition “mutual information $I$ signifies the amount of information obtained about one random variable by observing another.” and its mathematical definition to Appendix B, where we use the definition to derive Eq. (13) from Eq. (12). Furthermore, in Section 5.3.1, which focuses on the mutual information experiment, we now include a pointer to Appendix D.2.6 for detailed information on the estimation methods used.
>
>
> ## W1 (Appendix B), Q1
> We firstly appreciate the Reviewer SF1W’s pointing out that the proof in Appendix B is contributing. As Reviewer SF1W’s feedback, similar statements for $I(\boldsymbol{h}, r)$ cannot be derived. Appendix B proves that minimizing $L_\mathrm{dict}$ increases the lower bound on $I(\boldsymbol{h},y)$. With the inequality used in [2],  $I(\boldsymbol{h}, r) <=  I(\boldsymbol{h}, \boldsymbol{x}) - I(\boldsymbol{h}, y)$,  $I(\boldsymbol{h}, r)$ will increase if $I(\boldsymbol{h},\boldsymbol{x})$ increase more than $I(\boldsymbol{h},y)$ increases. Likewise, $I(\boldsymbol{h}, r)$ will decrease if $I(\boldsymbol{h},y)$ increase more than $I(\boldsymbol{h},\boldsymbol{x})$ increases. Thus, to make a claim on $I(\boldsymbol{h},r)$, we should make a deductive claim on how $L_\mathrm{dict}$ affects $I(\boldsymbol{h},\boldsymbol{x})$.
>
>
> Theoretically, making such a claim appears implausible, given that $L_{dict}$ pertains solely to the relationship between $\boldsymbol{t}_z$ and $\boldsymbol{h}$, and does not extend to the relationship between $\boldsymbol{h}$ and $\boldsymbol{x}$. Instead, Figure 4 empirically demonstrates a trend that $I(\boldsymbol{h}, \boldsymbol{x})$ progressively decreases as layers progress, resulting in the decline of $I(\boldsymbol{h},r)$.
>
>
> In other words, we focused on the empirical aspect of $I(\boldsymbol{h}, r)$, demonstrating that our loss function is more effective in reducing $I(\boldsymbol{h}, r)$ compared to conventional contrastive loss in the absence of auxiliary networks $f_\phi$. Although the proof in Appendix B is vital to our paper, given the considerations mentioned earlier, please consider that we are unable to relocate it due to its considerable length.
>
>
> However, we already know $I(r, y)=0$ by the definition of task irrelevant variable $r$: "any random variable that affects the observed data $x$, but is not informative
> to the task we are trying to solve" such that $I(r, y) = 0$ [1,2]. Intuitively speaking, observing task irrelevant variable $r$ gives no insight about task-relevant variable $y$, and vice versa.
>
>
>
>
> ## W2
> Thank Reviewer SF1W for highlighting the issue with the variable $z$. To reduce confusion, we have revised the notation for auxiliary network output from $\boldsymbol{z}=f_\phi(\boldsymbol{h})$ to $\boldsymbol{a}=f_\phi(\boldsymbol{h})$. Additionally, to ensure consistency, we have unified the class index as 'z' in Eq  (2) and Appendix B, and changed $y\in\mathbb{N}$ to $y\in\{1,...,Z\}$ on page 3.
>
>
> ## Q2
> Many thanks for this interesting suggestion. The outcomes from your suggestion are indeed meaningful. As depicted in Figure 15, the t-SNE visualization of label embeddings post-training on CIFAR-100 (with 20 super-labels and 100 sub-labels) illustrates a noteworthy pattern. Labels within the same super-label group tend to cluster together. Furthermore, labels from different super-label groups are observed to be in close proximity if they share semantic similarities (e.g., the embedding for 'forest' is closer to that of 'trees' compared to other embeddings from the 'large outdoor scene' category.)”. We have incorporated a discussion of these findings into Section 5.3 of our paper.
>
> ## Minors
> Thanks for pointing out our mistakes on citations. We have revised the citation.
>
> [1] Alessandro Achille and Stefano Soatto. Emergence of invariance and disentanglement in deep
> representations. The Journal of Machine Learning Research, 19(1):1947–1980, 2018.
>
>
> [2] Yulin Wang, Zanlin Ni, Shiji Song, Le Yang, and Gao Huang. Revisiting locally supervised learning:
> an alternative to end-to-end training. In International Conference on Learning Representations,
> 2020

---

> > ### Comment · Reviewer_SF1W · 2023-11-20
> > **Response to Authors**
> >
> > Dear authors,
> >
> > Thank you for integrating my suggestions into the paper. Given the details added to the paper (e.g., visualization of label embeddings) and the correction of presentation issues, all my concerns were addressed. As a result, I am updating my score from 6: Marginally above the acceptance threshold to __8: accept, good paper__.

---

> > > ### Author Response · Authors · 2023-11-21
> > > **Response to Reviewer SF1W**
> > >
> > > We extend our sincere gratitude to Reviewer SF1W for the reconsideration and reassessment of our work.

---

### Official Review · Reviewer_3KwZ · 2023-10-30

**Soundness:** 3 good
**Presentation:** 3 good
**Contribution:** 4 excellent
**Rating:** 8
**Confidence:** 4

**Summary:**

The paper offers a new algorithm for Forward Learning that is based on intermediate layer features being close to adaptive label embeddings. This work shows comparison to various FL techniques, and shows that the proposed technique is beneficial in terms of performance, and memory usage.

**Strengths:**

The paper builds on intuition existing in many previous works (see Questions section for potential missing literature) to create a new FL algorithm. The work is presented well, and the experiments look convincing. Further, the method is simplistic, intuitive, and in my view has great potential in future works. The authors also provided code for reproducibility, which is great.

**Weaknesses:**

I REALLY enjoyed reading the paper, and I think other than some missing literature and visual comments, the paper is very good.
One possible comment is that the architectures used are a not exactly SOTA - I would love to see this done with ResNets, ViT, RegNets and so forth - this could enhance the paper greatly.

**Questions:**

**Questions**:
1. It is interesting that the authors chose to have one embedding per class for all layers. One could think that different layers can achieve different levels of clustering and therefore both the embeddings and the loss weight on them can be different (see for example [2]). Another alternative to this would be to learn an MLP from the features to the embeddings, rather than using average pooling. I wonder if the authors tried this direction.
2. Another question is wether the authors considered having more than one embedding per class, given that there may be variability within each class. - this could then be some hyperparameter: number of embs per class


**Possible missing literature**:
1. I think the paper relates very much to the literature of neural collapse in intermediate layers. It would be great to see the authors mention these works and make the connection. [1-4, 6]

2. There have been previous works that use intermediate layer losses to encourage class label clustering in the BP setting [2, 5].


[1] Papyan, V., Han, X. Y., & Donoho, D. L. (2020). Prevalence of neural collapse during the terminal phase of deep learning training. Proceedings of the National Academy of Sciences, 117(40), 24652-24663. https://doi.org/10.1073/pnas.2015509117

[2] Ben-Shaul, I. &amp; Dekel, S.. (2022). Nearest Class-Center Simplification through Intermediate Layers. <i>Proceedings of Topological, Algebraic, and Geometric Learning Workshops 2022</i>, in <i>Proceedings of Machine Learning Research</i> 196:37-47 Available from https://proceedings.mlr.press/v196/ben-shaul22a.html.

[3] Ben-Shaul, I., Shwartz-Ziv, R., Galanti, T., Dekel, S., & LeCun, Y.  Reverse Engineering Self-Supervised Learning. In Proceedings of the Thirty-seventh Conference on Neural Information Processing Systems (NeurIPS 2023).

[4]  Rangamani, A., Lindegaard, M., Galanti, T. &amp; Poggio, T.A.. (2023). Feature learning in deep classifiers through Intermediate Neural Collapse. <i>Proceedings of the 40th International Conference on Machine Learning</i>, in <i>Proceedings of Machine Learning Research</i> 202:28729-28745 Available from https://proceedings.mlr.press/v202/rangamani23a.html.

[5] Gamaleldin F. Elsayed, Dilip Krishnan, Hossein Mobahi, Kevin Regan, and Samy Bengio. Large margin deep networks for classification. In NeurIPS, 2018.

[6] Galanti, T., Galanti, L., & Ben-Shaul, I. (2023). Comparative Generalization Bounds for Deep Neural Networks. Transactions on Machine Learning Research, (ISSN 2835-8856).

---

> ### Author Response · Authors · 2023-11-20
> **Response to Reviewer 3KwZ (1/4)**
>
> ## W1
>
>
> We thank Reviewer 3KwZ for the suggestion. Because we are more familiar with MLP-Mixer than RegNet, we test $L_\mathrm{dict}$ with ResNet-32, ViT, and MLP-Mixer in Appendix E. These architectures are composed of multiple modules, with each one acting as a functional unit. For example, in ViT, the multi-head attention (MHA) and MLP cooperate as a single unit (module); MHA is responsible for capturing attention, and the MLP is tasked with processing it. We maintain that these modules are integral to their respective architectures and should remain coupled, as decoupling them would disrupt their designed functionality. Consequently, in our experimental setup, we applied $L_{dict}$ at a module level, incorporating module-wise BP. However, we did not employ module-wise auxiliary networks used in LL. Moreover, we compare our results with $L_\mathrm{contrast}$ (as defined in Eq (1)), which does use such networks.
>
>
> |         | ResNet-32        |         |             | ViT             |         |             | MLP-Mixer      |         |             |
> |---------|------------------|---------|-------------|-----------------|---------|-------------|----------------|---------|-------------|
> |         | Errs.            | Δθ      | Memory      | Errs.           | Δθ      | Memory      | Errs.          | Δθ      | Memory      |
> | BP      | 6.701            | 0       | 3179 MiB    | 16.25           | 0       | 5300 MiB    | 17.23          | 0       | 6361 MiB    |
> | $L_{contrast}$ | 32.71          | 73.6K   | 1617 MiB    | 32.95          | 394K   | 1354 MiB    | 31.67         | 394K   | 1468 MiB    |
> | $L_{dict}$     | 25.19          | 5.12K   | 1617 MiB    | 32.63          | 5.12K  | 1348 MiB    | 23.51         | 5.12K  | 1445 MiB    |
>
>
> The results, which are compiled in Table 8, indicate that our approach is not only more parameter and memory efficient but also yields superior results compared to $L_\mathrm{contrast}$ across all tested architectures.
>
>
>
> ## Q1 (One embedding per class for all layers).
> We express our sincere appreciation for Reviewer 3KwZ’s valuable suggestions. Following the advice, we have applied and evaluated the layer-wise dictionary $\boldsymbol{D}^Z_l$ across datasets.
> | Method     | MNIST | F-MNIST | CIFAR-10 | CIFAR-100 | SVHN  | STL-10 |
> |------------|-------|---------|----------|-----------|-------|--------|
> | DC-FL      | 0.33  | 5.52    | 8.64     | 31.75     | 2.19  | 22.87  |
> | DC-FL-LD   | 0.34  | 5.50    | 8.45     | 31.64     | 2.19  | 22.59  |
>
> The results indicate that the layer-wise approach outperforms a single dictionary overall. Moreover, Reviewer 3KwZ’s recommendation presents an effective method for parallel training contexts where each layer could be processed on a separate GPU.
>
>
> ## Q1 (Replacing average pooling with MLP)
> We again convey our deep gratitude for  Reviewer 3KwZ’s insightful recommendations concerning alternatives to pooling. In an experiment based on this suggestion, we mapped $\boldsymbol{t}_z$ to $\boldsymbol{t}_z^l$ using a layer-wise fully connected layer $f_P$, which act as the auxiliary networks for $\boldsymbol{t}_z$. The results, as displayed in Table 9, were unexpected: $f_P$ led to a decline in performance for both $\boldsymbol{D}^Z$ and $\boldsymbol{D}^Z_l$. Thanks again to Reviewer 3KwZ’s suggestion. This experiment provides an ablation study on $pool_l$.
>
>
>
> |       | CIFAR-10     |           | CIFAR-100   |           |
> |-------|--------------|-----------|-------------|-----------|
> |       | Errs.        | Memory    | Errs.       | Memory    |
> | $\boldsymbol{D}^Z_l$ | 11.70        | 770 MiB    | 33.15      | 785 MiB    |
> | $\boldsymbol{D}^Z$   | 12.47        | 753 MiB    | 36.08      | 770 MiB    |
> | $ pool_l $   | 8.64         | 747 MiB    | 31.75      | 751 MiB    |

---

> ### Author Response · Authors · 2023-11-20
> **Response to Reviewer 3KwZ (2/4)**
>
> ## Q2
> We express our deepest gratitude for  Reviewer 3KwZ’s valuable insights. Following  this suggestion, we have conducted an experiment to explore the concept of multiple embeddings per class, specifically in a scenario where we can more precisely discern variations within each class. Given that CIFAR-100 is organized into 20 super-classes, each comprising 5 sub-labels, we know exactly how each super-class label (super-label) varies within the 5 sub-labels, thereby obtaining 5 sub-labels per super-label. We tested models trained on CIFAR-100 to make inferences on 20 super-labels, using two methods, 1) assigning the super-label based on the closest proximity of the local feature $\boldsymbol{h}$ to any sub-label embeddings within the super-label (“Super”); 2) averaging the sub-label embeddings to create a unified super-label embedding, which is then used to predict the closest super-label during inference (“Mean”).
> |       | FC    | Conv  | VGG   |
> |-------|-------|-------|-------|
> | E2E   | 56.65 | 40.72 | 23.42 |
> | Naive | 65.85 | 49.78 | 31.75 |
> | Super | 60.95 | 37.56 | 21.84 |
> | Mean  | 60.98 | 48.61 | 32.26 |
>
>
> The results, presented in Table 7 in the revised script, reveal that the "Super" method outperforms the others, while the "Mean" method does not improve upon the baseline approach (predicting sub-label's super-label as the prediction). We speculate that this is due to the presence of outlier sub-labels. For instance, as demonstrated in Figure 7 in the revised script, the "forest" sub-label is more closely aligned with the "trees" super-label rather than its own "large natural outdoor scene" group, skewing the average embedding for the super-label and causing a bias. Additionally, our results indicate that the model learns semantic relationships more effectively than dataset-defined taxonomies. For example, "chimpanzee" is categorized under
> "large omnivores and herbivores" alongside "elephant, cattle, camel, and kangaroo," yet it more intuitively aligns with "people," as illustrated in Figure 7.

---

> ### Author Response · Authors · 2023-11-20
> **Response to Reviewer 3KwZ (3/4)**
>
> ## In Relation to Neural Collapse
> We extend our deepest thanks to Reviewer 3KwZ for insightful recommendations of significant literature. Our method aligns closely with the neural collapse phenomenon, particularly NCC or NC4 [1]. As [2,3,4] suggest, neural collapse also emerges in the intermediate layers. By minimizing $\mathbb{E}[L_{dict}(\boldsymbol{h},\boldsymbol{D}_Z)]$, our approach actively encourages the alignment between the average of local features with label $z$ and label embedding vectors $\boldsymbol{t}_z$, akin to the motivation of local losses used in [2,5].
>
>
> However, our work is different from these studies in its exploration of locally decoupled contexts. Existing works [2,3,4,5] focus on local features within an end-to-end BP framework, whereas we investigate these features in scenarios where each layer or module functions independently. In this perspective, every local feature can be considered as the “final feature before the classifier”, which is where neural collapse emerges. In fact, as detailed in Appendix C, our methodology is capable of making layer-wise predictions. Figure 9 in the revised script demonstrates an increase in accuracy as we progress through the layers, resonating with observations made in [3].
>
>
> This unique perspective allows us to uncover and understand the effects of neural collapse in a different and possibly more nuanced computational environment. We've drawn connections to neural collapse in our revised manuscript in Appendix F, even though it wasn't the central focus of our original paper, to elucidate the relevance of our findings within the broader landscape of neural network research.

---

> ### Author Response · Authors · 2023-11-20
> **Response to Reviewer 3KwZ (4/4)**
>
> [1] Papyan, V., Han, X. Y., & Donoho, D. L. (2020). Prevalence of neural collapse during the terminal phase of deep learning training. Proceedings of the National Academy of Sciences, 117(40), 24652-24663. https://doi.org/10.1073/pnas.2015509117
>
>
> [2] Ben-Shaul, I. & Dekel, S.. (2022). Nearest Class-Center Simplification through Intermediate Layers. <i>Proceedings of Topological, Algebraic, and Geometric Learning Workshops 2022</i>, in <i>Proceedings of Machine Learning Research</i> 196:37-47
>
>
> [3] Ben-Shaul, I., Shwartz-Ziv, R., Galanti, T., Dekel, S., & LeCun, Y. Reverse Engineering Self-Supervised Learning. In Proceedings of the Thirty-seventh Conference on Neural Information Processing Systems (NeurIPS 2023).
>
>
> [4] Rangamani, A., Lindegaard, M., Galanti, T. & Poggio, T.A.. (2023). Feature learning in deep classifiers through Intermediate Neural Collapse. <i>Proceedings of the 40th International Conference on Machine Learning</i>, in <i>Proceedings of Machine Learning Research</i> 202:28729-28745 Available from https://proceedings.mlr.press/v202/rangamani23a.html.
>
>
> [5] Gamaleldin F. Elsayed, Dilip Krishnan, Hossein Mobahi, Kevin Regan, and Samy Bengio. Large margin deep networks for classification. In NeurIPS, 2018.

---

### Official Review · Reviewer_UJZ4 · 2023-11-02

**Soundness:** 2 fair
**Presentation:** 2 fair
**Contribution:** 2 fair
**Rating:** 6
**Confidence:** 3

**Summary:**

This work studies local training, that is training with local error signals so as to eliminate backprop, which is not plausible biologically. The proposed method is dictionary contrastive learning, built upon a prior work that proposed using (supervised) contrastive learning for local learning. The main difference from that prior work is using a embedding dictionary $t_1, \cdots, t_Z$ for each of the $Z$ classes, and aligning intermediate features with the average pooling of these embeddings (for dimension agreement).

**Strengths:**

1. I think the subject matter of this work is very interesting, and I can see why contrastive learning is relevant in this context, though I should point out that neither the problem nor contrastive learning for local learning is first proposed by this submission.
2. The authors conduct a number of experiments to show that the proposed method is better than previous local learning methods, which is good, though I would say that there is one very important baseline that the authors do not compare to (see below).

**Weaknesses:**

I am not an expert of local learning or forward learning, though I am very familiar with representation learning and contrastive learning. So I read the most relevant cited papers in this work, and from my understanding, this work is a follow-up work of Wang et al. [1]. And my understanding is that the main difference between this work and [1] is the introduction of $\lbrace t_1, \cdots, t_Z \rbrace$, the dictionary. So while conventional contrastive learning maximizes the similarity between a positive pair of samples' features, this work maximizes the similarity between a sample's feature and its corresponding $t_i$, and $t_i$ is also being updated during training. My following review is mostly based on the above understanding, and the comparison between this work and [1]. Please let me know if there is anything that I misunderstand.

The following are my major questions and concerns:
### 1. Regarding Sections 3 and 4
Sections 3 and 4, arguably the most important two sections of this work, are really confusing.
- Figure 1 compares "$L_{feat}$" with "$L_{contrast}$", and at first glance seems to suggest that contrastive learning is better than some other method called "feat", but this is not the case. Both methods minimize the contrastive loss, and the difference is that "feat" uses the intermediate outputs $h$ directly while "contrast" fits a network $\phi$ on top of it. So this figure does not show that contrastive learning is better, or "the performance of FF-based approaches falls short in comparison to InfoPro". Instead, it only shows that using a neural network $\phi$ is important. Thus, I am not quite sure what message the authors want to convey with Figure 1, as well as Figure 3.
- An important motivation of this work and LL/FL as a whole is the goal of removing backprop (BP), because BP is not biologically plausible and is not memory efficient. The abstract and intro of this submission make this point very clear. My question is: If you use a neural network $\phi$ in Eqn. (1), then how can you remove BP? Don't you use BP to train this $\phi$? It might be true that using a smaller network $\phi$ can save memory, but my point is BP is still there if you are using this $\phi$, unless you are updating this $\phi$ with local learning too (which I don't think is the case). Thus, I don't think the proposed method can be called forward learning (FL), which by definition should have no BP at all. And it is not very fair to compare the proposed method with FL methods.
- In Section 4, "Mapping labels to embedding vectors", the authors wrote "the label embeddings in the dictionary are updated at each step, making them a dynamic concept". But how are these $t_i$ updated? Are they updated by minimizing the loss $L_{dict}$? If this is the case, then this update is not local, because remember that the same $t_i$ are shared across all layers (which then go through pooling), so the update signals of $t_i$ come from all layers. Consequently, the layers cannot be learned in parallel as FL methods, because all layers need to jointly update $t_i$. I would say that it makes more sense to me if these $t_i$ are fixed, unless there are some imperative reasons that $t_i$ must be updated.
- In Eqn. (2), why there is an average over $K$? And why does there need to be $K$ feature maps for one class in the first place? Eqn. (2) can be simplified as $L_{dict} = -\frac{1}{N} \sum [ \text{log} \frac{\text{exp}( \langle h_n, t_+' \rangle )  }{ \sum \text{exp}( \langle h_n, t_i' \rangle ) } ]$, where $h_n$ is the average of $h_n^k$. So why not just use one $h_n$, but use K $h_n^k$ if they are equivalent?

### 2. Regarding the experiments
- My biggest concern with the experiments is that this work is built upon InfoPro [1], yet in the experiments it is not compared to InfoPro [1] at all. I feel that it is very likely that the method proposed in this work has a very similar performance to InfoPro.
- The experiments compare the proposed method with FL methods, which as I said is not very fair, because I don't think the proposed method is FL since it still uses backprop when updating $\phi$. InfoPro [1] did not claim their method to be FL. In fact, their paper did not mention FL at all.

[1] Wang et al., Revisiting locally supervised learning: an alternative to end-to-end training, ICLR 2021.

**Questions:**

Minor points:
- The authors wrote $K = H \times W$ in several places, which makes little sense to me. Why does an image need to have as many feature maps as the number of its pixels? And why does it need to have multiple feature maps in the first place?
- Can you explicitly write out the formula of $pool_l$? For example, what does it do when $C_h^l > C_D$? Does it do zero padding? And can you tell me what is the motivation of pooling, other than adjusting the dimensions?


**Summary:** Overall, though I do think that the idea of introducing a dictionary could be interesting, I don't think this work presents this idea very well and makes enough justification. The paper is really confusing at times. Moreover, I feel that even the authors themselves are confused sometimes, when they call their method FL which to my understanding really isn't, and use $K=HW$ features in Eqn. (2) which is equivalent to using a single feature. Thus, I recommend rejecting this submission.

**Rebuttal note:** Score changed from 3 to 6.

My suggestion to the authors would be to read Wang et al. [1] very carefully, which I did before reviewing this submission. Their motivation was to improve over end-to-end training, and they did not mention removing BP or forward learning at all. That's why it makes sense for their method to use a BP-updated neural network, as long as it is modularized. Also, given the great similarity between this work  and [1], InfoPro should be the most important baseline in your experiments.

---

> ### Author Response · Authors · 2023-11-20
> **Response to Reviewer UJZ4 (1/3)**
>
> # Regarding InfoPro, auxiliary networks
> We are truly grateful to Reviewer UJZ4 for insights and understanding. While Reviewer UJZ4 has overall grasped our work well, we would like to clarify one particular aspect: we did not utilize auxiliary networks $f_\phi$ in our approach. As Reviewer UJZ4 pointed out, LL uses module-wise auxiliary networks and hence has module-level BP to update auxiliary networks. Yet, FL does not use auxiliary networks, being free from BP. Therefore, we can say that DC-FL is a FL framework. Our motivating experiments (in Section 3) starts from what happens if there is no auxiliary network (i.e., FL) in InfoPro [1], comparing $L_\mathrm{feat}$ (auxiliary network-free InfoPro) to $L_{\mathrm{contrast}}$ (InfoPro). This observation (Figure 1) motivated the need for a novel solution that is entirely independent of $f_\phi$ while still delivering competitive performance.
>
>
> With the above-mentioned respective, though our proposed objective is quite similar to that of InfoPro in that both use contrastive loss, they are different from each other (LL vs. FL). Nonetheless, sincerely consenting with Reviewer UJZ4, we conducted a further comparison with InfoPro.
> | Method     | MNIST | F-MNIST | CIFAR-10 | CIFAR-100 | SVHN  | STL-10 |
> |------------|-------|---------|----------|-----------|-------|--------|
> | LL-contrec | *0.65 | *5.71   | *9.02    | *31.35    | *2.34 | *29.74 |
> | LL-cont    | *0.37 | *5.92   | *7.72    | *31.19    | *2.29 | *26.83 |
> | DC-FL      | 0.33  | 5.52    | 8.64     | 31.75     | 2.19  | 22.87  |
>
>
> | Method     | MNIST, F-MNIST |       | CIFAR-10, SVHN |       | CIFAR-100 |       | STL-10 |       |
> |------------|----------------|-------|----------------|-------|-----------|-------|--------|-------|
> |            | Δθ             | Memory Cost | Δθ             | Memory Cost | Δθ    | Memory Cost | Δθ    | Memory Cost |
> | LL-contrec | 1.15M          | 811 MiB     | 2.07M          | 1049 MiB     | 2.07M | 1050 MiB     | 2.07M | 5954 MiB    |
> | LL-cont    | 918K           | 695 MiB     | 1.84M          | 894 MiB      | 1.84M | 895 MiB      | 1.84M | 1846 MiB    |
> | DC-FL      | 5.12K          | 580 MiB     | 5.12K          | 747 MiB      | 51.2K| 751 MiB      | 5.12K | 1589 MiB    |
>
>
> |         | ResNet-32        |         |             | ViT             |         |             | MLP-Mixer      |         |             |
> |---------|------------------|---------|-------------|-----------------|---------|-------------|----------------|---------|-------------|
> |         | Errs.            | Δθ      | Memory      | Errs.           | Δθ      | Memory      | Errs.          | Δθ      | Memory      |
> | BP      | 6.701            | 0       | 3179 MiB    | 16.25           | 0       | 5300 MiB    | 17.23          | 0       | 6361 MiB    |
> | $L_{contrast}$ | 32.71          | 73.6K   | 1617 MiB    | 32.95          | 394K   | 1354 MiB    | 31.67         | 394K   | 1468 MiB    |
> | $L_{dict}$     | 25.19          | 5.12K   | 1617 MiB    | 32.63          | 5.12K  | 1348 MiB    | 23.51         | 5.12K  | 1445 MiB    |
>
>
>
> We have included the results in the following Table 3,4,8 in the revised script. Our method continues to show competitive performance against InfoPro (LL), while being more memory and parameter efficient. These additional findings have been incorporated into the revised manuscript.
>
> [1] Wang et al., Revisiting locally supervised learning: an alternative to end-to-end training, ICLR 2021.

---

> ### Author Response · Authors · 2023-11-20
> **Response to Reviewer UJZ4 (2/3)**
>
> ## Regarding Parallel Training
> Thank the Reviewer UJZ4 for pointing out the parallel training issue. We agree with Reviewer UJZ4 on the need to address the parallel training scenario. The version presented in the script updates the label embeddings sequentially offering a benefit in terms of memory efficiency (as the allocated memory is freed after each forward pass of the module). Additionally, layer-wise gradients for $\boldsymbol{t}_z$ can be synchronized by averaging across all layers, just as DistributedDataParallel in Pytorch averages batch-wise gradients across devices to synchronize (allowing parallel training). However, we admit that updating from the average gradients may cause training instability because label embeddings receive concurrent error signals from all layers. To address this issue, Reviewer UJZ4 ‘s suggestion (fixed embeddings) can be a solution. However, our experiment illustrated in Figure 5 already exhibits that
> updating $\boldsymbol{t}_z$ is better. Thus, we propose two alternative strategies more suitable for parallel training: (i) updating the label embeddings only at the last intermediate layer (DC-FL-O), and (ii) employing independent label embeddings for each layer (DC-FL-LD). Both versions still demonstrate competitive performance and
> memory/parameter efficiency.
> | Method     | MNIST | F-MNIST | CIFAR-10 | CIFAR-100 | SVHN  | STL-10 |
> |------------|-------|---------|----------|-----------|-------|--------|
> | DC-FL      | 0.33  | 5.52    | 8.64     | 31.75     | 2.19  | 22.87  |
> | DC-FL-O    | 0.32  | 5.39    | 8.68     | 34.58     | 2.18  | 23.56  |
> | DC-FL-LD   | 0.34  | 5.50    | 8.45     | 31.64     | 2.19  | 22.59  |
>
> | Method     | MNIST, F-MNIST |       | CIFAR-10, SVHN |       | CIFAR-100 |       | STL-10 |       |
> |------------|----------------|-------|----------------|-------|-----------|-------|--------|-------|
> |            | Δθ             | Memory Cost | Δθ             | Memory Cost | Δθ    | Memory Cost | Δθ    | Memory Cost |
> | DC-FL-LD   | 21.8K          | 581 MiB     | 21.8K          | 749 MiB      | 218K | 766 MiB      | 21.8K | 1593 MiB    |
> | DC-FL      | 5.12K          | 580 MiB     | 5.12K          | 747 MiB      | 51.2K| 751 MiB      | 5.12K | 1589 MiB    |
>
> We have added this in Tables 3 and 4 in the revised script.
>
> ## Regarding $\boldsymbol{h}_n^k$
>
>
> We agree your advice that $\frac{1}{K}\sum^K_{k=1}\langle\boldsymbol{h}_n^k, \boldsymbol{t} \rangle$ is mathematically equivalent to $\langle \boldsymbol{\bar{h}}_n, \boldsymbol{t}\rangle$, where $\boldsymbol{\bar{h}}_n=\frac{1}{K}\sum^K_{k=1}\boldsymbol{h}_n^k$. In fact, we used the simplified expression (what Reviewer UJZ4 suggested) for the mathematical proof in Appendix B. However, please consider that ‘averaging feature vectors first’ could degrade performance due to more severe loss in floating-point precision, which is typically more pronounced when handling smaller numbers. Averaging the dot products, which generally result in higher values, is less prone to precision loss. Conversely, averaging the feature vectors before the dot product can lead to greater precision loss, given that the values in feature vectors are usually smaller. Our intention was to accurately represent the implemented version of the loss. However, considering the simplicity of the expression Reviewer UJZ4 highlighted, we have revised the notation in Eqn. (2) and included a footnote to address the potential loss of floating-point precision.

---

> ### Author Response · Authors · 2023-11-20
> **Response to Reviewer UJZ4 (3/3)**
>
> ## Clarification on K
>
>
> Thank Reviewer UJZ4  for the advice regarding $K$. To address this, we want to first clarify that $K$ in our study differs from "$K$" in InfoPro [1]. "$K$" in [1] refers to the number of local modules, each containing multiple layers, whereas $K$ in our paper signifies the number of feature vectors. At the l-th layer, a feature map $\boldsymbol{h}_{map}^l \in \mathbb{R}^{C_h^l\times H_l\times W_l}$ holds $K_l$ (i.e., $H_l\times W_l$) feature vectors, each of dimension $C_h^l$. We adopted $K$ for two main reasons: i) to standardize the format of local features between fully connected (FC) and convolutional (Conv) layers; and ii) to establish the notation for averaging feature vectors over $K$.
>
> For FC layers, local output is a flat vector $\boldsymbol{h_{flat}}^l \in \mathbb{R}^{C_{flat}^l}$. We can unify the expression for $\boldsymbol{h_{flat}}$ and $\boldsymbol{h_{map}}$, by considering $\boldsymbol{h_{flat}}$ as $\boldsymbol{h} \in \mathbb{R}^{C_h^l\times K_l}$, thereby setting $C_{flat}^l = C_h^lK_l$. Furthermore, our findings suggest that for FC layers, averaging over $K_l>1$ yields better results than using the flat vector $K_l=1$, as evidenced in Appendix K.
>
>
> ## Regarding $C_D<C_h^l$ and $pool_l$
>
>
> For a model with L layers, we make sure $C_D = \max\limits_lC_h^l$. When $C_D>C_h^l$, we use pooling to map $\boldsymbol{t}_z \in \mathbb{R}^{C_D}$ to $\boldsymbol{t}_z^l \in \mathbb{R}^{C_h^l}$.
>
>
> For $pool_l$, we use the average pooling with a padding of 0, a stride of $\lfloor\frac{C_D}{C^l_h}\rfloor$, and kernels of size $C_D - (C^l_h-1) \times \mathrm{stride}$ (torch.nn.AdaptiveAvgPool1d). We employed pooling as it is a simple, but effective method to reduce vector dimensions without resorting to BP. Appendix G reveals that pooling not only simplifies the process but also outperforms using a FC layer $f_P^l$ to map $\boldsymbol{t}_z$ to $\boldsymbol{t}^l_z$.

---

> > ### Comment · Reviewer_UJZ4 · 2023-11-20
> > **Reviewer Response**
> >
> > I thank the authors for their response and their revision.
> >
> > 1. I apologize for my earlier understanding. If I understand it correctly now, the goal of this work is to implement a variant of InfoPro but without BP, so that it is forward learning. The new tables are good.
> >
> > 2. Parallel training: In my original review, I didn't mean that parallel training is not implementable. My point was that since LL and FL have a biological motivation, having a global update signal does not seem plausible biologically. But as I said, I am not an expert in LL and FL so my understanding could be wrong. The new DC-FL-O and DC-FL-LD seem more biologically plausible to me.
> >
> > 3. $h _n^K$: The authors said that the reason for having multiple feature vectors is for numerical stability, which I understand. In that case, your general formulation should just have one feature vector, but then when really implementing this, you can mention that for numerical stability, you use multiple feature vectors. The new revision of the paper is good.
> >
> > 4. $K = H \times W$: If I understand it correctly, $K = H \times W$ is related to a specific model architecture. I still cannot see why there is a need to have so many feature vectors (the number of pixels can be really large), but maybe it is just for implementation convenience.
> >
> > 5. Pooling: Thank you. Your response makes sense.
> >
> > I am raising my score from 3 to 6. And after discussing with my fellow reviewers and AC, I might further raise my score.

---

> > > ### Author Response · Authors · 2023-11-21
> > > **Response to Reviewer UJZ4**
> > >
> > > We are grateful to Reviewer UJZ4 for dedicating time to thoroughly comprehend and reassess our work.
> > >
> > > ### 2
> > >
> > > Reviewer UJZ4 raised a valid concern regarding the parallels between our brain's processing and the architecture in question. Our brain handles information in a highly parallel manner, so a global structure can receive update signals in parallel. However, as Reviewer UJZ4 has insightfully pointed out, a global signal perfectly synchronized across layers seems less plausible. Accordingly, DC-FL-O and DC-FL-LD indeed offer a more biologically plausible approach for asynchronous parallelism. Conceptually, our work draws inspiration from the template/prototype matching theories of cognitive science, which align with the idea of neural networks using template/prototypes for pattern recognition.
> > >
> > > ### 4
> > > It is correct that $K = H \times W$ is essentially architecture-specific, used for the sake of implementation convenience.

---

### Author Response · Authors · 2023-11-20
**Upload of Revised Script**

We have revised our script with text color red (Reviewer UJZ4), orange (Reviewer 3KwZ), and blue (Reviewer SF1W).

---

### Meta-Review · Area_Chair_VEH7 · 2023-12-02

**Metareview:**

**Summary:** The work focuses on the study of local training, specifically training with local error signals instead of backpropagation (i.e., forward learning).  The proposal introduces the use of a dictionary learned through contrastive learning, distinguishing itself by learning an embedding dictionary for each class.  The approach utilizes forward learning and compares multiple techniques, showing performance and memory usage improvements.  The main motivation behind this loss is to reduce mutual information between feature representations and eliminate task-irrelevant information.  The reviewers agree that the paper introduces novel and important ideas that will be of interest for the ML community.

**Strengths:** The proposed forward learning algorithm, based on a dictionary of label embeddings, exhibits strong performance in practical settings.  It demonstrates comparable or superior results compared to existing state-of-the-art approaches.  Notably, the algorithm stands out for its relatively lightweight nature, offering efficiency advantages over other methods in this domain.

**Weaknesses:** 'The reviewers generally find the paper interesting and appreciate the authors' experiments and contribution, especially in proposing a new training objective based on a dictionary of label embeddings. While some reviewers note that the problem and contrastive learning for local learning are not first proposed by this submission, the method is presented well and is considered simplistic and intuitive with great potential for future works. Reviewers also commend the authors for providing code for reproducibility and find the resulting algorithm to be comparable or even better than existing state-of-the-art while being relatively light-weight in comparison with existing approaches.

**Justification For Why Not Higher Score:**

The paper has notable contributions and has been praised by the reviewers. However, there are weaknesses that need to be addressed, particularly concerning the limited experiments conducted. Despite these concerns, the reviewers recommend the paper for spotlight, but not for oral presentation.

**Justification For Why Not Lower Score:**

The reviewers unanimously acknowledge the paper's significant contributions despite the identified shortcomings, indicating that it should not be rejected. Additionally, although the reviewers initially suggested a poster or spotlight presentation, given their high praise, I believe it deserves a higher placement. Furthermore, as the paper delves into a different paradigm of neural network learning, it will be intriguing for the community to explore these novel techniques as an inspiration for future endeavors.

---

### Decision · Program_Chairs · 2024-01-16

Accept (spotlight)